# Singular Value Fine-tuning: Few-shot Segmentation requires Few-parameters Fine-tuning

**Yanpeng Sun**[1]*, **Qiang Chen**[2]*, **Xiangyu He**[3]*, **Jian Wang**[2], **Haocheng Feng**[2]
**Junyu Han**[2], **Errui Ding**[2], **Jian Cheng**[3], **Zechao Li**[1]†, **Jingdong Wang**[2]
[1]School of Computer Science and Engineering, Nanjing University of Science and Technology
[2]Baidu VIS
[3]NLPR, Institute of Automation, Chinese Academy of Sciences

## Abstract

Freezing the pre-trained backbone has become a standard paradigm to avoid over-fitting in few-shot segmentation. In this paper, we rethink the paradigm and explore a new regime: *fine-tuning a small part of parameters in the backbone*. We present a solution to overcome the overfitting problem, leading to better model generalization on learning novel classes. Our method decomposes backbone parameters into three successive matrices via the Singular Value Decomposition (SVD), then *only fine-tunes the singular values* and keeps others frozen. The above design allows the model to adjust feature representations on novel classes while maintaining semantic clues within the pre-trained backbone. We evaluate our *Singular Value Fine-tuning (SVF)* approach on various few-shot segmentation methods with different backbones. We achieve state-of-the-art results on both Pascal-$5^i$ and COCO-$20^i$ across 1-shot and 5-shot settings. Hopefully, this simple baseline will encourage researchers to rethink the role of backbone fine-tuning in few-shot settings. The source code and models will be available at `https://github.com/syp2ysy/SVF`.

## 1 Introduction

Benefiting from the large amounts of annotated data, deep learning has achieved noticeable improvements in the field of semantic segmentation [44, 17, 34]. In contrast, their performances dramatically degrade when novel classes arrive or label data is insufficient. Thus, few-shot segmentation (FSS) [22, 49] was proposed to address these challenges. In FSS, one needs to segment novel class objects in query images given only a few densely-annotated samples (*i.e.*, support images and support masks).[3] Due to the extremely limited data in FSS, over-fitting has become a critical problem that needs to be carefully handled.

One feasible solution is to restrict the model's learning capacity so that it can not overfit the small dataset. Most recent works [43, 47, 30, 33] follow this idea by freezing the pre-trained backbone. Then, different feature fusion methods and prototypes are introduced to enhance the generalization ability. Although this paradigm has achieved promising results, it is still suboptimal to directly adopt parameters pre-trained on image classification to image segmentation. The semantic clues contained in the pre-trained backbone can be irrelated to objects shown in support images, bringing unexpected obstacles to segmenting novel class objects in FSS.

In this paper, we rethink the paradigm of freezing the pre-trained backbone and show that fine-tuning *a small part of parameters in the backbone* is free from overfitting, leading to better model generalization in learning novel classes. Our method is illustrated in Figure 1(b). First, to find such a

---

*Equal Contribution.

†Corresponding author.

[3]In this paper, we consider two settings of few-shot segmentation: 1-shot and 5-shot, which contain only one and five support images and masks, respectively.

36th Conference on Neural Information Processing Systems (NeurIPS 2022).

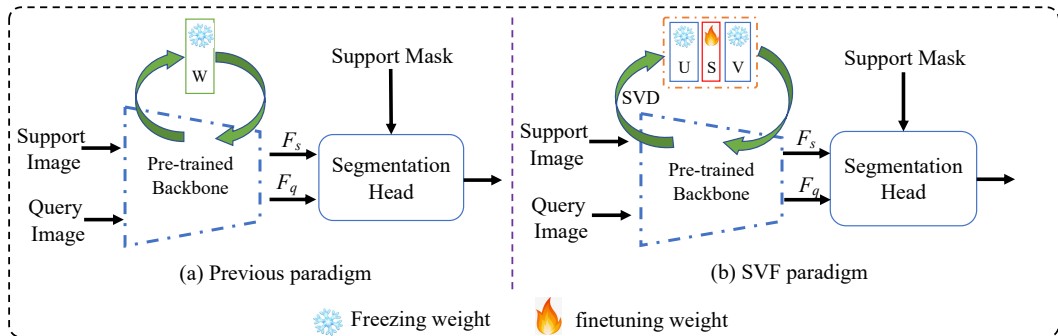

Figure 1: Previous paradigm *vs.* SVF. (a) Previous paradigm introduces different segmentation heads based on the frozen pre-trained backbone. (b) SVF uses SVD to decompose the pre-trained parameters into three consecutive matrices, then only fine-tune the singular values and keep others frozen. Compared to the previous paradigm, SVF shows that fine-tuning a small part of parameters in the backbone is invulnerable to over-fitting, leading to better model generalization in learning novel classes.

small part of parameters for fine-tuning, we decompose pre-trained parameters into three successive matrices via the Singular Value Decomposition (SVD). Second, we then *fine-tune the singular value matrices* and keep others frozen. The above design, called *Singular Value Fine-tuning (SVF)*, follows two principles: (i) maintaining rich semantic clues in the pre-trained backbone and (ii) adjusting feature map representations when learning to segment novel classes.

We evaluate our SVF on two few-shot segmentation benchmarks, Pascal-$5^i$ and COCO-$20^i$. Extensive experiments show that SVF is invulnerable to overfitting and works well with various FSS methods using different backbones. It is significantly better than the freezing backbone counterpart, leading to new state-of-the-art results on both Pascal-$5^i$ and COCO-$20^i$. Moreover, we provide quantitative and qualitative analyses on how singular values change during fine-tuning. Results show that SVF helps models focus more on the objects to be segmented instead of the noisy background. Our experiments highlight that proper backbone fine-tuning consistently outperforms backbone freezing on several leading methods. We hope our simple method will encourage researchers to rethink the role of backbone fine-tuning for few-shot segmentation.

## 2 Related Work

### 2.1 Few-shot Segmentation

The purpose of few-shot segmentation is to segment the unseen class in query image with a few densely-annotated samples. In this task, a semantically rich representation and a nice matching approach have a particularly large impact on the results. Therefore, mainstream methods [43, 40, 32, 5] focus on obtaining excellent prototypes from support images, and obtaining accurate segmentation results by improving the quality of prototype features. CANet [45], PFENet [35] and PANet [37] filters class irrelevant information by global average pooling to obtain foreground and background prototypes. ASGNet [16] pointed out that increasing the number of prototypes can further improve the segmentation results. CyCTR [47] believes that pixel-level features in support image are important for segmentation tasks, and proposes to use pixel-level prototypes to predict query images. On the other hand, some methods focus on designing better matching methods to improve segmentation performance. SG-One [49] uses cosine similarity to match prototype and query feature for segmentation results. CANet [45] proposes an additive alignment module to iteratively refine the network output. HSNet [23] exploits neighborhood consensus to disambiguate semantics by analyzing patterns of local neighborhoods in matching tensors. In addition to the above work, BAM [15] utilizes the segmentation results of the base class to guide the generation of unseen classes, and achieves SOTA results. However, the above methods are all based on backbone freeze, and freezing backbone not only reduces the representational ability of the model, but also does not fit distribution to data better. Unlike previous work, in this paper we focus on the prospect of fine-tuning backbone in FSS. Therefore, instead of proposing a new model, we adopt the classic PFENet [35] and BAM [15] as our baselines. Our SVF enables these methods to further improve segmentation results.

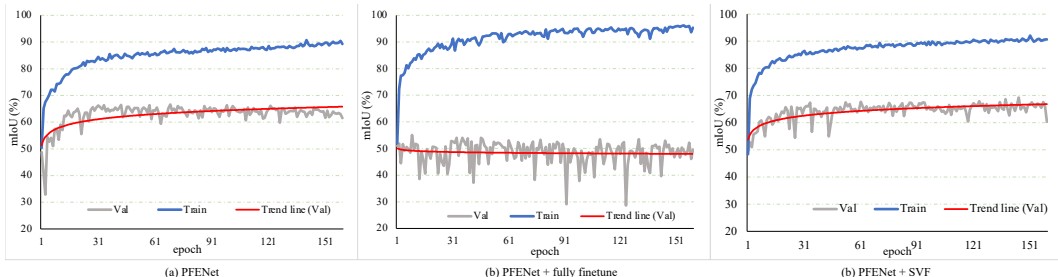

Figure 2: The mIoU curve of PFENet [35] with different fine-tune strategies on Pascal-$5^i$ Fold-0. (a) is the result of freezing backbone, (b) and (c) represent the result of fine-tuning the entire backbone and SVF, respectively. Compared with the direct fine-tuning (b), SVF not only avoids the overfitting problem, but also brings positive results.

## 2.2 Backbone Fine-tuning

Fine-tuning backbone in downstream tasks has become a common approach in deep learning. The initial breakthroughs in vision tasks [10, 13, 18] were achieved by fine-tuning classification networks based ImageNet pre-trained weight, such as R-CNN [7] for detection and FCN [20] for segmentation. However, the direct application of fine-tuing the entire backbone in few-shot scenarios will lead to over-fitting of the model [30]. Therefore, fine-tuning part parameters of the backbone in few-shot learning may avoid model over-fitting. P-Transfer [30] utilizes NAS to search parameters of backbone that require fine-tune in few-shot classification tasks. However, this method is very complicated and cannot be directly applied to small sample segmentation. And some works [14] borrow the idea of prompt [36, 31, 39] in NLP to fine-tuning part parameters in the visual transformer [6]. The above methods are proposed in a transformer-based model, but most few-shot segmentation models use CNN-based backbones. Applying prompt-based methods to various few-shot segmentation methods may need further adjustments. Different from the above methods, our SVF borrows the commonly used SVD [1] in model compression and constructs [4, 50] a novel part fine-tune method for few-shot segmentation task. In addition, some approaches [28, 26, 25] introduce highly constrained subset of parameters to fine-tuning. However, these methods are not applied on few-shot segmentation task.

## 3 From Freezing Backbone to Singular Value Fine-tuning

In this section, we start with the preliminaries on the few-shot segmentation (FSS) setting. Then, we revisit the overfitting problem in FSS when fine-tuning the backbone in Section 3.1. In Section 3.2, we propose a novel *Singular Value Fine-tuning (SVF)* method for FSS instead of freezing the pre-trained backbone as proposed in previous methods. Section 3.3 provides a discussion on the differences between SVF and other fine-tuning methods.

**FSS Setup.**  Few-shot segmentation (FSS) aims to segment novel class objects given only a few densely-annotated samples. In this task, datasets are split into the training set ($\mathbb{D}_{train}$) with base classes ($\mathbb{C}_{train}$) and the testing set ($\mathbb{D}_{test}$) with novel classes ($\mathbb{C}_{test}$), where $\mathbb{C}_{train} \cap \mathbb{C}_{test} = \emptyset$. Following previous works, we adopt episode training. Each episode consists of $k$ support images and one query image to construct a $k$-shot segmentation task ($k = 1$ or $k = 5$ in this paper). Then, FSS methods are trained with episodes to segment novel class objects in the query image given the knowledge of $k$ support images and support masks.

## 3.1 Revisiting Model Overfitting in FSS

As presented in Section 1, model overfitting is a critical problem in extremely limited data scenarios (1-shot and 5-shot), especially when the model has large amounts of learnable parameters. We validate this problem and design experiments with a typical FSS method, PFENet [35]. Figure 2 shows that as model training moves on, fine-tuning backbone leads to better performance on the training dataset while it does not improve results on the validation set. It is a typical overfitting phenomenon. In contrast, freezing backbone can achieve steady improvements on the validation set during training.Therefore, existing methods in FSS turn to freezing the pre-trained backbone to avoid the overfitting problem.

Although this strategy has achieved promising results, it is clear that directly adopting an ImageNet pre-trained backbone to image segmentation can be suboptimal. One need to extract the most related semantic clues within the backbone instead of involving too much noise coming from the irrelevant

categories learned from upstream tasks. In light of this, we rethink the paradigm of freezing the pre-trained backbone and try to find a new solution to the overfitting problem.

## 3.2 Singular Value Fine-tuning

According to the analysis above, fine-tuning all parameters in the backbone can be unsatisfactory. One feasible solution is to restrict the backbone's learning capacity so that it is suitable for the few-shot circumstance. Instead of freezing the whole pre-trained backbone as in previous works, we consider exploring a new regime, which is *only fine-tuning a small part of the parameter in the backbone*.

However, it is nontrivial to find such a small part of parameters to be fine-tuned in the backbone. Simply splitting the backbone's parameters into learnable and freezing ones results in negative results. Table 5 and Table 6 show the inferior performances no matter splitted by layers or convolution types. We attribute this to the adjustment of the backbone, making the model *biased towards base class objects* shown in the training set, yet leads to worse model generalization in segmenting novel classes. Figure 3 also provides evidences of the overfitting problem. Given that the backbone is pre-trained on a

---

**Algorithm 1** Pseudocode of SVF in Python style

```
# Input: Conv2d with weight matrix W, Input feature X
# Output: Output feature Y

# Previous 3x3 Conv :
# Y = F.Conv2d(W, X, kernel=(3,3))

# SVF :
U, S, V = svd(W) # decompose weights by SVD

U.requires_grad = False # freeze Conv_U
V.requires_grad = False # freeze Conv_V

Y = F.Conv2d(V, X, kernel=(3,3)) # a new 3x3 conv
Y = Y.mul(S) # reconstruct a new affine layer
Y = F.Conv2d(U, Y, kernel=(1,1)) # a new 1x1 conv
```

---

large-scale dataset with a classification task, it contains rich semantic clues but it is suboptimal to adopt for a segmentation task directly. We deliver two principles for finding a small part of fine-tuning parameters in the backbone: (i) maintaining rich semantic clues in the pre-trained backbone and (ii) adjusting feature map representations when learning to segment novel classes.

To fulfill the above goal, we resort to model compression methods in this paper. They are designed to approximate the original pre-trained model with fewer parameters, and also follow the above two principles. Among these methods, low-rank decomposition is a common technique to achieve model compression. It first splits the model weights into multiple subspaces and then compresses each subspace by shrinking its rank. We follow this direction and decompose the backbone parameters into subspaces via the Singular Value Decomposition (SVD). However, we do not shrink subspaces' ranks since our target is to find a small part of parameters to be fine-tuned instead of model compression.

In detail, for a convolution layer with $C_i$ input channels, $C_o$ output channels, and a kernel size of $K \times K$ in the pre-trained backbone, we first fold its weight tensor $\mathbf{W} \in \mathbb{R}^{C_o \times C_i \times K \times K}$ into a matrix $\mathbf{W}' \in \mathbb{R}^{C_o \times C_i K^2}$, then decompose the obtained matrix by applying SVD with full-rank in subspaces (rank $R = \min(C_o, C_i K^2)$). Thus,

$$\mathbf{W}' = \mathbf{U}\mathbf{S}\mathbf{V}^T, \text{where } \mathbf{U} \in \mathbb{R}^{C_o \times R}, \mathbf{S} \in \mathbb{R}^{R \times R}, \text{and } \mathbf{V}^T \in \mathbb{R}^{R \times C_i K^2}. \tag{1}$$

The obtained pair of matrices $\mathbf{V}^T$ and $\mathbf{U}$ construct two new convolution layers, and $\mathbf{S}$ is a diagonal matrix with singular values on the diagonal. Then back to convolutions, the results of the Equation 1 corresponds to three successive layers: a $R \times C_i \times K \times K$ convolution layer, a scaling layer, and followed by a $C_o \times R \times 1 \times 1$ convolution layer (Algorithm 1 further provides a pseudo-code for SVF). We thus split each convolution layer in the pre-trained backbone into three functionalities: (i) decouple the semantic clues into a subspace with rank $R$, (ii) re-weight the semantic clues with singular values for the given task, and (iii) project the re-weighted clues back to the original space. Based on the interpretation above, we propose to fine-tune the scaling layer, which is Singular Value Fine-tuning (SVF). SVF does not erase the semantic clues contained in the pre-trained backbone, but it re-weights the representations to help adjust the model for new segmentation tasks. As SVF restricts the learnable parameters to only singular values, which are extremely few ($0.25\%$) compared with the parameters in the whole backbone, SVF is less vulnerable to overfitting and shows better generalization capacity in learning novel classes (shown in Table 5, Table 6, Figure 2, and Figure 3).

## 3.3 Discussion on Fine-tuning Methods

Fine-tuning pre-trained backbones is a promising way to achieve state-of-the-art results on downstream vision tasks. Many fine-tuning methods have been introduced to transfer pre-trained backbone's knowledge, such as full-model fine-tuning [10, 46], task-specific fine-tuning (freezing the

Table 1: Performance on Pascal-$5^i$[29] in terms of mIoU for 1-shot and 5-shot segmentation. The best mean results are show in **bold**. † indicates that images from training set containing the novel class on test set were removed.

| Method | backbone | 1-shot | | | | | 5-shot | | | | |
|---|---|---|---|---|---|---|---|---|---|---|---|
| | | Fold-0 | Fold-1 | Fold-2 | Fold-3 | Mean | Fold-0 | Fold-1 | Fold-2 | Fold-3 | Mean |
| baseline† | VGG16 | 57.48 | 66.72 | 62.66 | 53.72 | 60.15 | 62.98 | 70.57 | 68.62 | 59.60 | 65.44 |
| baseline†+SVF | | 63.07 | 68.40 | 65.81 | 54.28 | **62.89**$_{(+2.74)}$ | 68.52 | 72.15 | 69.08 | 63.59 | **68.34**$_{(+2.90)}$ |
| PFENet†[35] | | 61.91 | 70.34 | 63.77 | 57.38 | 63.35 | 67.73 | 72.82 | 69.31 | 67.59 | 69.36 |
| PFENet†+SVF | | 63.43 | 71.40 | 64.18 | 58.30 | **64.33**$_{(+0.98)}$ | 69.11 | 73.67 | 69.13 | 67.30 | **69.80**$_{(+0.44)}$ |
| BAM†[15] | | 63.18 | 70.77 | 66.14 | 57.53 | 64.41 | 67.36 | 73.05 | 70.61 | 64.00 | 68.76 |
| BAM†+SVF | | 64.09 | 71.07 | 66.79 | 57.54 | **64.87**$_{(+0.46)}$ | 67.75 | 74.11 | 70.99 | 63.57 | **69.11**$_{(+0.35)}$ |
| baseline† | ResNet50 | 65.60 | 70.28 | 64.12 | 60.27 | 65.07 | 69.89 | 74.16 | 67.87 | 65.73 | 69.41 |
| baseline†+SVF | | 67.42 | 71.57 | 67.99 | 61.57 | **67.14**$_{(+2.07)}$ | 70.37 | 75.06 | 71.08 | 69.16 | **71.42**$_{(+2.01)}$ |
| baseline | | 66.36 | 69.22 | 57.64 | 58.73 | 62.99 | 70.75 | 72.92 | 58.86 | 65.56 | 67.02 |
| baseline + SVF | | 66.88 | 70.84 | 62.33 | 60.63 | **65.17**$_{(+2.18)}$ | 71.49 | 74.04 | 59.38 | 67.43 | **68.09**$_{(+1.07)}$ |
| PFENet†[35] | | 66.61 | 72.55 | 65.33 | 60.91 | 66.35 | 70.93 | 75.32 | 69.60 | 68.96 | 71.20 |
| PFENet†+SVF | | 69.27 | 73.55 | 67.49 | 62.30 | **68.15**$_{(+1.80)}$ | 71.82 | 74.92 | 70.97 | 69.58 | **71.82**$_{(+0.62)}$ |
| PFENet | | 67.06 | 71.61 | 55.21 | 59.46 | 63.34 | 72.11 | 73.67 | 61.61 | 67.50 | 68.72 |
| PFENet + SVF | | 68.31 | 71.99 | 56.25 | 61.82 | **64.59**$_{(+1.25)}$ | 72.09 | 73.99 | 63.58 | 70.03 | **69.92**$_{(+1.20)}$ |
| BAM†[15] | | 68.97 | 73.59 | 67.55 | 61.13 | 67.81 | 70.59 | 75.05 | 70.79 | 67.20 | 70.91 |
| BAM†+SVF | | 69.38 | 74.51 | 68.80 | 63.09 | **68.95**$_{(+1.14)}$ | 72.05 | 76.17 | 71.97 | 68.91 | **72.28**$_{(+1.37)}$ |
| BAM | | 68.37 | 72.05 | 57.55 | 60.38 | 64.59 | 70.72 | 74.21 | 63.58 | 66.18 | 68.67 |
| BAM + SVF | | 68.17 | 72.86 | 57.77 | 62.04 | **65.21**$_{(+0.62)}$ | 72.30 | 74.43 | 65.16 | 69.43 | **70.33**$_{(+1.66)}$ |

backbone)[9, 41], residual adapter [12, 2, 38], and bias tuning [48, 8]. We compare our SVF with these methods. As presented in previous sections, methods like full-model fine-tuning may not be suitable with extremely limited data, and task-specific fine-tuning can not provide adjustment to the representations in the backbone for downstream tasks. Moreover, methods like residual adapter and bias tuning need prior knowledge to model structure or weights. SVF does not have the above limitations in the few-shot segmentation task.

Recently, a new fine-tuning method named Vision Prompt Tuning (VPT) [14] has been proposed to fine-tune vision transformers. It introduces a small number of trainable parameters in the input space while keeping the backbone frozen. From this perspective, our SVF also introduces a small number of trainable parameters but in the singular value space. In SVF, the learned singular value diagonal matrix $\mathbf{S}$ can be formulated as a product of a frozen matrix $\mathbf{S}_{frozen}$ and a trainable matrix $\mathbf{S}_{trainable}$, which is $\mathbf{S} = \mathbf{S}_{frozen}\mathbf{S}_{trainable}$. We give a detailed explanation of this perspective in the Appendix.

## 4 Experiments

We conduct experiments on Pascal-$5^i$ [29] and COCO-$20^i$ [24] to discuss fine-tuning approach in FSS. In this section, we first introduce the used representative method and implementation details. Then we discuss the impact of different fine-tune methods on the FSS model, and finally verify the effectiveness and versatility of the proposed fine-tune method.

### 4.1 Setting

**Datasets.** Experiments are conducted on Pascal-$5^i$[29] and COCO-$20^i$ [24]. Following the previous work [23, 35, 29], we separate all classes in both datasets into 4 folds. For each fold, Pascal-$5^i$[29] has 15 classes used for training and 5 classes for test, COCO-$20^i$ [24] has 60 classes used for training and 20 classes for test. To verify the performance of the model, we randomly sample 1000 query-support pairs in each fold. Following the BAM [15], we remove images from training set containing novel classes of test to prevent potential information leakage. We give a detailed explanation of this setting about train sets in the Appendix.

**Dataset Tricks:** The previously methods annotated novel classes in the training set as background during training step. It become a common paradigm in few-shot segmentation. However, based on BAM [15], we found a novel dataset trick to improve the performance of FSS models. It simply removes images from the training set that contain the novel classes. For fair comparison with BAM, we use this trick in our experiments. However, we know that previously methods does not use this trick. Therefore, we present the experimental results with and without dataset trick in Table 1, and more detailed fair comparison results in Appendix. Here, we hope that researchers can make fair comparison under the same setting.

Table 2: Performance on COCO-20$^i$ [24] in terms of mIoU for 1-shot and 5-shot segmentation. The best mean results are show in **bold**. $^\dagger$ indicates that images from training set containing the novel class on test set were removed.

| Method | Backbone | 1-shot | | | | | 5-shot | | | | |
|---|---|---|---|---|---|---|---|---|---|---|---|
| | | Fold-0 | Fold-1 | Fold-2 | Fold-3 | Mean | Fold-0 | Fold-1 | Fold-2 | Fold-3 | Mean |
| baseline$^\dagger$ | VGG-16 | 37.56 | 37.73 | 38.78 | 35.66 | 37.43 | 43.07 | 49.42 | 44.38 | 46.38 | 45.81 |
| baseline$^\dagger$+SVF | | 39.32 | 39.64 | 38.63 | 35.45 | **38.26**$_{(+0.83)}$ | 46.48 | 50.72 | 45.79 | 45.63 | **47.16**$_{(+1.35)}$ |
| PFENet [35] | | 35.40 | 38.10 | 36.80 | 34.70 | 36.30 | 38.20 | 42.50 | 41.80 | 38.90 | 40.40 |
| PFENet$^\dagger$ [35] | | 41.03 | 44.22 | 43.74 | 38.90 | 41.97 | 48.66 | 48.26 | 45.49 | 51.02 | 48.36 |
| PFENet$^\dagger$+SVF | | 42.68 | 44.90 | 42.60 | 38.79 | **42.24**$_{(+0.27)}$ | 49.02 | 53.71 | 47.59 | 47.63 | **49.49**$_{(+1.13)}$ |
| BAM$^\dagger$ [15] | | 38.96 | 47.04 | 46.41 | 41.57 | 43.50 | 47.02 | 52.62 | 48.59 | 49.11 | **49.34** |
| BAM$^\dagger$+SVF | | 40.21 | 46.62 | 46.23 | 41.97 | **43.76**$_{(+0.26)}$ | 45.05 | 53.59 | 48.35 | 49.28 | 49.07$_{(-0.27)}$ |
| baseline$^\dagger$ | ResNet-50 | 38.91 | 46.07 | 42.67 | 39.71 | 41.84 | 50.35 | 56.78 | 49.61 | 50.96 | 51.93 |
| baseline$^\dagger$+SVF | | 44.22 | 46.38 | 42.65 | 41.65 | **43.72**$_{(+1.88)}$ | 51.47 | 57.48 | 50.33 | 52.29 | **52.89**$_{(+1.93)}$ |
| PFENet$^\dagger$ [35] | | 44.93 | 50.32 | 44.68 | 44.26 | 46.05 | 52.29 | 59.34 | 51.50 | 53.53 | 54.17 |
| PFENet$^\dagger$+SVF | | 46.88 | 50.86 | 47.69 | 46.64 | **48.02**$_{(+1.97)}$ | 52.72 | 58.14 | 52.52 | 54.15 | **54.38**$_{(+0.21)}$ |
| BAM$^\dagger$ [15] | | 43.41 | 50.59 | 47.49 | 43.42 | 46.23 | 49.26 | 54.20 | 51.63 | 49.55 | 51.16 |
| BAM$^\dagger$+SVF | | 46.87 | 53.80 | 48.43 | 44.78 | **48.47**$_{(+2.24)}$ | 52.25 | 57.83 | 51.97 | 53.41 | **53.87**$_{(+2.71)}$ |

Table 3: Performance on Pascal-5$^i$[29] in terms of FB-IoU for 1-shot and 5-shot segmentation.

| Method | backbone | FB-IoU (%) | |
|---|---|---|---|
| | | 1-shot | 5-shot |
| baseline$^\dagger$ | ResNet-50 | 77.11 | 80.56 |
| baseline$^\dagger$+SVF | | **78.86** | **82.66** |
| PFENet$^\dagger$ | | 77.35 | 82.30 |
| PFENet$^\dagger$+SVF | | **79.07** | **82.77** |
| BAM$^\dagger$ | | **81.10** | 82.18 |
| BAM$^\dagger$+SVF | | 80.13 | **83.17** |

Table 4: Ablation study of BN on Pascal-5$^i$ under 1-shot setting. ✓represents fine-tuning this feature space. The best mean results are show in **bold**.

| Method | BN | S | Mean |
|---|---|---|---|
| baseline$^\dagger$ | - | - | 65.07 |
| | ✓ | | 63.12$_{(-1.95)}$ |
| | ✓ | ✓ | 64.20$_{(-0.87)}$ |
| | | ✓ | **67.14**$_{(+2.07)}$ |

**Methods.** To quickly verify the effectiveness of SVF, we propose a simple baseline method. Then, two representative methods are used to verify the generality of SVF. Next, we briefly introduce these three methods. Meanwhile, for fair comparison, we unify the three methods into the same framework.

- **Baseline**: We replace FEM module on PFENet [35] with ASPP module to get the baseline method. The baseline only sets main loss, without adding auxiliary loss.
- **PFENet [35]**: As a classic method in FSS, it proposed the Prior Guided Feature Enrichment Network, which has a huge impact on the subsequent FSS methods.
- **BAM [15]**: As the state-of-the-art method in FSS, it represents the most cutting-edge results in FSS.

**Implementation details.** In our experiments, we use VGG-16 [3] and ResNet-50 [11] as the backbone network and initialize it with ImageNet [27] pre-trained weights. Due to the particularity of BAM, we use the initialization weights provided by author, and SVF does not finetuning base branch in BAM. We use SGD optimizer with cosine Learning rate decay [21], the learning rate 0.015 and the random seed 321 when fine-tuning backbone. We keep original settings for the model without fine-tuning. All models are trained 200 epochs on Pascal-5$^i$ with batch size 8 and trained 50 epochs on COCO-20$^i$ with batch size 8. Image is resized to $473 \times 473$ on Pascal-5$^i$ and $641 \times 641$ on COCO-20$^i$. Following [19, 41, 42] , we adopt the mean intersection over union (mIoU) and foreground-background IoU (FB-IoU) as our evaluation metric. Since the middle-level features and high-level features are used in all FSS model, we set the SVF to fine-tuning the parameters of layers 2, 3, and 4 only. All model runs on four NVIDIA A100 GPUs.

## 4.2 Comparison with State-of-the-Art

In this section, the effectiveness of SVF is validated on three most representative methods. For fair comparison, we rerun all methods with unified framework. Then, we compare different methods with singular value fine-tuning and freeze backbone. The experimental results on Pascal-5$^i$ are in Table 1. It can be seen that the performance of different methods has been significantly improved after SVF. When the backbone is VGG-16, the SOTA method BAM improves miou by $0.46$ and $0.35$ in 1-shot

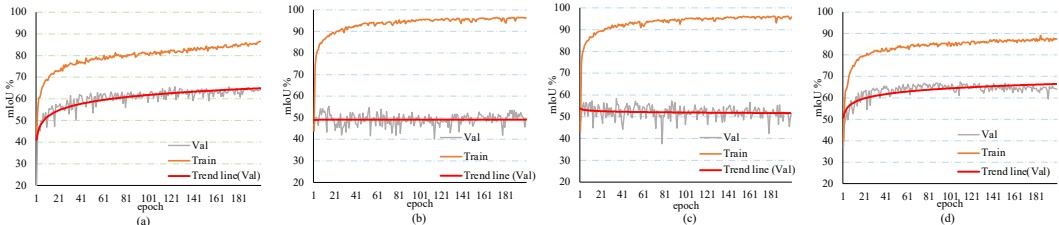

Figure 3: The mIoU curve of baseline with different finetune strategies on Pascal-$5^i$ Fold-0. (a) represent the results of baseline with freezing backbone, (b) represent directly fine-tuning layers 2, 3 and 4, (c) represent fine-tuning all $1 \times 1$ convolution layer in backbone, and (d) represent the proposed method SVF.

Table 5: Comparative experiment with fine-tuning different layer of backbone on Pascal-$5^i$.

| Method | layer | Mean |
|---|---|---|
| baseline$^\dagger$ | - | 65.07 |
| +fully fine-tune | 1, 2, 3, 4 | $60.90_{(-4.17)}$ |
| + part fine-tune | 2, 3, 4 | $61.15_{(-3.92)}$ |
| | 3, 4 | $61.08_{(-3.99)}$ |
| | 4 | $60.58_{(-4.49)}$ |
| +SVF | 2, 3, 4 | $\mathbf{67.14}_{(+2.07)}$ |

Table 6: Comparative experiment with fine-tuning different convolutional layer of backbone on Pascal-$5^i$.

| Method | layer | $3 \times 3$ | $1 \times 1$ | Mean |
|---|---|---|---|---|
| baseline$^\dagger$ | - | - | - | 65.07 |
| +part fine-tune | 2, 3, 4 | ✓ | ✓ | $61.15_{(-3.92)}$ |
| | 2, 3, 4 | ✓ | | $61.86_{(-3.21)}$ |
| | 2, 3, 4 | | ✓ | $61.62_{(-3.45)}$ |
| +SVF | 2, 3, 4 | - | - | $67.14_{(+2.07)}$ |

and 5-shot respectively after singular value fine-tuning. However, when the backbone is ResNet-50, SVF improves BAM by $1.14$ and $1.37$ mIoU on 1-shot and 5-shot, respectively. It shows that SVF bring better performance in deeper backbone. Meanwhile, Table 2 shows the effectiveness of SVF on more complex dataset COCO-$20^i$. Exspecially, SVF improves the performance of BAM by $2.24$ and $2.71$ mIoU on 1-shot and 5-shot. Furthermore, Table 3 shows the comparison results with FB-IoU on Pascal-$5^i$. Our SVF can also improve the performance of model. This experiment proves that SVF not only achieve state-of-the-art results, but also is a general method in FSS. In addition, the results (without $^\dagger$) in Table 1 prove that the dataset trick can indeed improve the performance of FSS model. It also shows that whether or not the dataset tricks is used does not affect the effectiveness of SVF. And we give more comparative results in the Appendix.

### 4.3  Ablation Study

To verify the effectiveness of SVF, we conduct a series of ablation study in this section. We use the baseline method to conduct ablation study on Pascal-$5^i$ 1-shot setting with ResNet-50 as the backbone network. Furthermore, we give the ablation study about hyperparameter in the Appendix.

**Batch Normalization (BN):** In Table 4, we test the effect of BN on SVF. In the case of only fine-tuning the BN layer, the baseline will greatly reduce the performance. Next, we test SVF (fine-tune subspace **S**) on the baseline without fine-tuning BN. The results show that SVF achieves the best performance. Finally, we test the performance of the baseline method when fine-tuning subspace **S** and BN simultaneously. The results show that fine-tuning parameters of BN layer can cause performance of baseline to degrade. Therefore, we freeze the parameters of BN layer when using SVF.

**Traditional fine-tune methods:** In this part, we conduct experiments to verify the impact of traditional fine-tuning methods on the FSS model. Traditional fine-tuning methods can be divided into fully fine-tune and part fine-tune. The fully fine-tune method means to fine-tuning all the parameters in the backbone. The part fine-tune methods means to fine-tuning part parameters in the backbone, which includes layer-based and convolution-based fine-tune methods. In table 5, we conduct quantitative experiments with fully and layer-based fine-tune on baseline method. The results show that fully fine-tune brings negative results to baseline method. Meanwhile, we find that the negative results of fully fine-tuning method are mitigated as the number of fine-tuning layers is reduced. However, these methods do not have a positive impact on the baseline. In table 6, we conduct quantitative experiments on convolution-based fine-tune methods. For fair comparison, we only fine-tuning convolutions of 2, 3 and 4 layers. The results show that only fine-tuning $3 \times 3$

Table 7: Ablation study of SVF fine-tuning different subspace on Pascal-$5^i$.

| Method | U | S | V | Mean |
|---|---|---|---|---|
| | ✓ | | | 61.09 |
| | | ✓ | | **67.14** |
| | | | ✓ | 60.88 |
| baseline† | ✓ | ✓ | | 61.57 |
| | | ✓ | ✓ | 60.42 |
| | ✓ | | ✓ | 60.02 |
| | ✓ | ✓ | ✓ | 61.24 |

Table 8: Ablation study of SVF fine-tuning different layer on Pascal-$5^i$. The best results are show in **bold**.

| Method | layer | Mean |
|---|---|---|
| | 4 | 66.21 |
| baseline† + SVF | 3, 4 | **67.20** |
| | 2, 3, 4 | 67.14 |
| | 1, 2, 3, 4 | 67.12 |

Table 9: Ablation study of different ways of changing semantic cues in weights on Pascal-$5^i$ 1-shot.

| Method | Expression of weight | Fine-tune param | Mean |
|---|---|---|---|
| | $W$ | - | 65.07 |
| | $S'W$ | $S'$ | 63.52 |
| | $WS'$ | $S'$ | 64.62 |
| baseline | $RS'R^TW$ | $S'$ | 43.36 |
| | $USV^T$ | $S$ | **67.14** |
| | $URSR^TV^T$ | - | 29.31 |
| | $URSR^TV^T$ | $S$ | 30.26 |

Table 10: Compare with parameter-efficient tuning methods on Pascal-$5^i$ 1-shot.

| Method | fine-tune method | Mean |
|---|---|---|
| | freeze backbone | 65.07 |
| | fully fine-tune | 60.90 |
| baseline | SVF | 67.14 |
| | Adapter | 20.71 |
| | bias tuning | 62.93 |

convolution or $1 \times 1$ convolution can further improve the performance of layer-based fine-tune methods. It show that traditional fine-tune methods cannot bring positive results. However, SVF brings positive results to the baseline method. The success of SVF proves that traditional fine-tune method destroys the rich semantic clues in the pre-trained backbone. In Figure 3, we compare the mIoU curves of the training and test sets in different fine-tune methods. The results show that part fine-tune method also produces the over-fitting problem. It proves that disrupting the rich semantic cues in pre-trained backbone will lead to model over-fitting, reducing model generalization. However, SVF solves the over-fitting problem without destroying semantic clues in pre-trained weight. And, it brings a new perspective for fine-tuning backbone.

**Fine-tuning which subspace:** To verify the influence of different sub-spaces on SVF, we conduct experiments on the subspace after SVD decomposition. The results are shown in Table 7. We find that only fine-tuning $S$ subspace brings positive results. Either fine-tuning the $U$ or $V$ subspace returns negative results. It shows that $U$ and $V$ contain rich semantic information in pre-trained weight after SVF. In other words, directly changing the feature distribution of the $U$ or $V$ subspace reduces the generalization ability of the model. To verify the above point, we test the performance of fine-tuning different subspace combinations. The results confirm that changing the distribution of $U$ or $V$ spaces brings negative results. The subspace $S$ represents the weight distribution of different semantic cues. Therefore, fine-tuning the subspace $S$ does not change the semantic cues of pre-trained weights. Meanwhile, adjusting the weights of different semantic cues enables model to better perform downstream tasks.

**Fine-tuning which layers:** Since the FSS model directly uses feature maps of layers 2, 3, and 4, we initially fine-tuning the subspace $S$ of layers 2, 3 and 4. However, this setting is unreasonable. To verify which layer $S$ have a greater impact on baseline, we conducted experiments on SVF under different layer combinations. The results are shown in Table 8. It can be seen that fine-tuning layers 3 and 4 achieves the best performance, while only fine-tuning the layer 4 achieves the lowest performance. It shows that semantic clues in layer3 are the most important for FSS. Next we discuss the reasons why SVF can achieve better performance by visualizing semantic cues in layer3.

**Compare with other parameter-efficient tuning methods:** Unlike SVF, the purpose of parameter-efficient tuning methods is to obtain performance similar with fully fine-tune by fine-tuning a small number of parameters. To verify the superiority of SVF over parameter-efficient tuning methods in FSS, we compare SVF with adapter [12] and bias tuning on Pascal-$5^i$ with the 1-shot setting. The details for adapter and bias tuning are given below:

- **Adapter:** Adapter is proposed in transformer-based models. When applying it into CNN-based backbone (ResNet), we make simple adjustments. We follow [12] to build the adapter structures and add them after the stages in the ResNet.
- **Bias Tuning:** In the ResNet backbone, the convolution layers do not contain bias term. The bias terms that can be used for tuning is the ones in BN layers. We fine-tune the bias terms in all BN layers in this method.

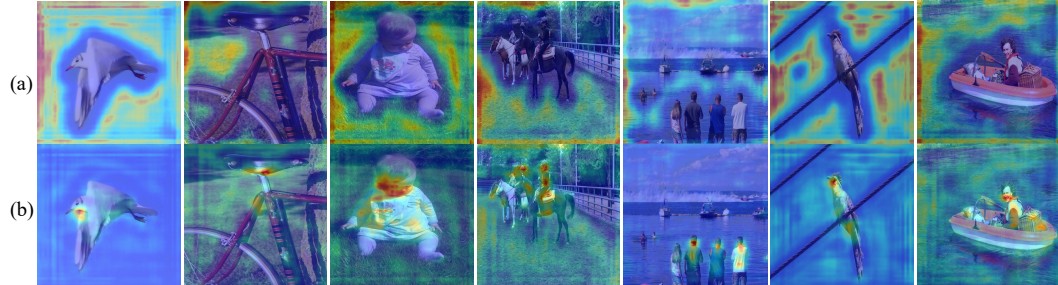

Figure 4: The visualization of segmentation cues with the largest variation in singular values from the last $3 \times 3$ convolution in layer 3. (a) represents segmentation clues of subspace $\mathbf{U}$ with the largest singular value reduction, (b) represents segmentation clues of subspace $\mathbf{U}$ with the largest singular value growth.

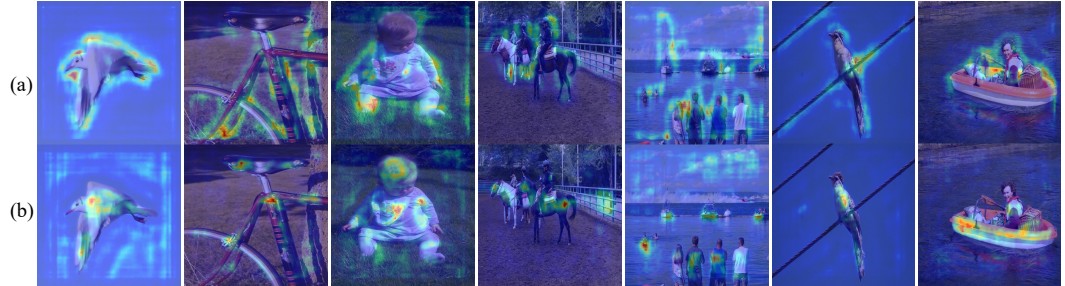

Figure 5: The visualization of segmentation cues with the largest variation in singular values from the last $1 \times 1$ convolution in layer 3. (a) represents segmentation clues of subspace $\mathbf{U}$ with the largest singular value reduction, (b) represents segmentation clues of subspace $\mathbf{U}$ with the largest singular value growth.

The experimental results are given in the table 10. It shows that SVF outperform Adapter and Bias Tuning by large margins. Moreover, we find that the introduction of Adapter will directly lead to over-fitting, while Bias Tuning reduces performance of the baseline model.

### 4.4 Discussion on Why SVF Works

The larger singular value in subspace $\mathbf{S}$, the more important semantic cues in subspace $\mathbf{U}$ and $\mathbf{V}$. We first focus on the changes of singular values during fine-tuning based on the initial distribution of $\mathbf{S}$. In Figure 6, we visualize the variation of Top-30 singular values in pre-trained weights. It can be seen that the singular values of either $1 \times 1$ or $3 \times 3$ convolution change dramatically after fine-tuning. Next, we visualize the semantic cues of subspace $\mathbf{U}$ with the largest variation

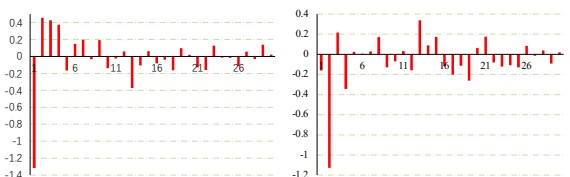

Figure 6: Statistics chart about the changes of initial Top-30 largest singular values of the last $1 \times 1$ and $3 \times 3$ convolution layer in layer3 after fine-tuning.

in singular values. The results are shown in Figure 4 and Figure 5. We only visualize the semantic cues where the singular value grows and decreases the most for a better view. Notice that semantic cues of decreasing singular values tend to focus on background regions. The semantic cues of increasing singular values always focus on foreground regions. The background-focused semantic cues in pre-trained backbone will damage the performance of FSS model. Since the original distribution of semantic cues in pre-trained backbone is not suitable for downstream tasks, SVF brings positive results to FSS model by increasing the weight of foreground cues and reducing the weight of background cues. It is also important to keep the semantic cues unchanged during fine-tuning. Overall, dynamically adjusting the weight of each cue without changing the semantic representation is the key to the success of SVF.

To verify that changes in the singular value space do not affect the semantic information in pre-trained weight, we conduct an interesting experiment to intentionally changing semantic cues in weights. In Table 9, We compare different approaches, including introducing a small number of

training parameters $S'$, and introducing a random rotation matrix $R$. It can be seen that changing the semantic cues in the weights negatively affects the FSS model (with or without fine-tuning a small number of parameters). Experimental results demonstrate that fine-tuning the singular value space is non-destructive (without destroy semantic cues). We give a detailed analysis about *why SVF work* in the Appendix.

### 4.5 Broader Impact

In this paper, we prove that freeze backbone is not the only paradigm in few-shot segmentation, fine-tune backbone is feasible. Meanwhile, we explore a new mechanism to redistribute the weights of different semantic cues without changing the semantic cues. As a new perspective of few-shot segmentation, it exposes the influence of pre-trained backbone on few-shot segmentation. Moreover, this mechanism not only works on few-shot, but also may be effective when fine-tune very large pre-trained models. This greatly reduces the cost of fine-tuning large models on downstream tasks.

### 4.6 Limitations

Although the above experiments demonstrate the power of SVF, it still has some limitations. For instance, SVF introduces a small number of learning parameters, but the occupancy rate of memory resources is high during training process. Using SVF in ResNet-50 will occupy 16G video memory per image in COCO-$20^i$ 5-shot setting. Furthermore, SVF increase a small amount of training time compared with freeze backbone.

## 5 Conclusion

In this paper, we rethink the paradigm of freezing backbone in FSS and propose a new paradigm Singular Value Fine-tuning (SVF) for fine-tuning backbone. Firstly, SVF decompose pre-trained parameters into three subspaces by SVD, and then only fine-tune the singular value. Our SVF dynamically adjusts the weights of different semantic cues without changing the rich semantic cues in pre-trained backbone. We evaluate the effectiveness of SVF on two commonly used benchmarks, Pascal-$5^i$ and COCO-$20^i$. Extensive experiments prove that SVF as a new perspective to avoid overfitting and significantly improve the performance of various FSS methods. As a new paradigm of finetune, we will extend it to a variety of vision tasks in the future.

**Acknowledge** This work was partially supported by the National Natural Science Foundation of China (Grant No. U20B2064 and U21B2043).

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
