# Singular Value Fine-tuning: Few-shot Segmentation requires Few-parameters Fine-tuning – Supplementary Material

**Yanpeng Sun**[1]\*, **Qiang Chen**[2]\*, **Xiangyu He**[3]\*, **Jian Wang**[2], **Haocheng Feng**[2]
**Junyu Han**[2], **Errui Ding**[2], **Jian Cheng**[3], **Zechao Li**[1]†, **Jingdong Wang**[2]
[1]School of Computer Science and Engineering, Nanjing University of Science and Technology
[2]Baidu VIS
[3]NLPR, Institute of Automation, Chinese Academy of Sciences

# Appendix

## A   More details

**Training Strategy:** Different from the training strategy of previous methods, we set the learning rate to 0.015 and use an SGD optimizer with cosine learning rate decay when fine-tuning the backbone. Therefore, we compared the impact of different training strategies on benchmark datasets. As shown in Table 1, the new training strategy does not affect the performance of FSS models. Therefore, *different training strategies are NOT the key to the success of SVF*.

Table 1: Compare with different training strategy on Pascal-$5^i$ training set in terms of mIoU for 1-shot segmentation.

| Method | Backbone | Training Strategy | 1-shot | | | | |
|---|---|---|---|---|---|---|---|
| | | | Fold-0 | Fold-1 | Fold-2 | Fold-3 | Mean |
| baseline | ResNet50 | original | 65.60 | 70.28 | 64.12 | 60.27 | 65.07 |
| baseline | | ours | 64.95 | 69.75 | 65.91 | 59.59 | 65.05 |
| PFENet (9) | | original | 66.61 | 72.55 | 65.33 | 60.91 | 66.35 |
| PFENet (9) | | ours | 65.58 | 72.49 | 66.12 | 60.30 | 66.12 |
| BAM (3) | | original | 68.97 | 73.59 | 67.55 | 61.13 | 67.81 |
| BAM (3) | | ours | 68.43 | 73.66 | 67.98 | 61.63 | 67.93 |

Table 2: Ablation study on the training trick.

| Method | Backbone | Training Trick | 1-shot | | | | |
|---|---|---|---|---|---|---|---|
| | | | Fold-0 | Fold-1 | Fold-2 | Fold-3 | Mean |
| baseline | ResNet50 | w/o | 66.36 | 69.22 | 57.64 | 58.73 | 62.99 |
| baseline | | w | 65.60 | 70.28 | 64.12 | 60.27 | 65.07 |
| PFENet (9) | | w/o | 67.06 | 71.61 | 55.21 | 59.46 | 63.34 |
| PFENet (9) | | w | 66.61 | 72.55 | 65.33 | 60.91 | 66.35 |
| CyCTR (15) | | w/o | 67.80 | 72.80 | 58.00 | 58.00 | 64.20 |
| CyCTR (15) | | w | 65.17 | 72.52 | 66.60 | 60.9 | 66.30 |
| BAM (3) | | w/o | 68.37 | 72.05 | 57.55 | 60.38 | 64.59 |
| BAM (3) | | w | 68.97 | 73.59 | 67.55 | 61.13 | 67.81 |

**Training Tricks:** Following the same setting of BAM (3), we remove some images containing novel classes of the test set from the training set. This is a novel trick in FSS to further improve

---

\*Equal Contribution.
†Corresponding author.

Table 3: compare with parameter-efficient tuning methods on Pascal-$5^i$ 1-shot.

| Method | fine-tune method | Fold-0 | Fold-1 | Fold-2 | Fold-3 | Mean |
|---|---|---|---|---|---|---|
| baseline | freeze backbone | 65.60 | 70.28 | 64.12 | 60.27 | 65.07 |
| | SVF | 67.42 | 71.57 | 67.99 | 61.57 | 67.14 |
| | Adapter | 18.41 | 20.21 | 26.62 | 17.62 | 20.71 |
| | bias tuning | 61.62 | 70.10 | 64.80 | 55.19 | 62.93 |

Table 4: Compare with different test image on COCO-$20^i$ in terms of mIoU for 1-shot segmentation.

| Method | backbone | test image | 1-shot | | | | |
|---|---|---|---|---|---|---|---|
| | | | Fold-0 | Fold-1 | Fold-2 | Fold-3 | Mean |
| baseline | ResNet50 | 1000 | 38.91 | 46.07 | 42.67 | 39.71 | 41.84 |
| baseline + SVF | | 1000 | 44.22 | 46.38 | 42.65 | 41.65 | 43.72 |
| baseline | | 4000 | 37.19 | 45.30 | 42.90 | 38.49 | 40.97 |
| baseline + SVF | | 4000 | 39.80 | 46.99 | 42.51 | 42.06 | 42.84 |
| baseline | | 5000 | 36.59 | 45.17 | 43.34 | 38.73 | 40.96 |
| baseline + SVF | | 5000 | 39.49 | 46.95 | 42.09 | 41.15 | 42.42 |

the performance. In Table 2, we compared the effect of this trick on FSS models. The results show that this trick brings 2.0 mIoU improvement over the original FSS model on average. Especially on Flod-2, the trend of improvement is very obvious. It proves that removing images with novel classes of the test set from the training set prevents potential information leakage.

**Test image of COCO-$20^i$:** We found that the number of test sets used in previous work was different when testing on COCO. For example, BAM (3), HSNet (6) were tested with 1000 images, yet *Yang* (12) was tested with 4000 images, and CyCTR (15) was tested with 5000 images. This is very detrimental to the development of the community. In Table 4, we compare the different number of test images on COCO-$20^i$ to observe changes in model performance. The experimental results show that as the number of test images increases, the performance of the baseline shows a downward trend. Therefore, we call on researchers to use the same training samples for a fair comparison. Meanwhile, SVF brings positive results in different numbers of test sets. It again shows the effectiveness of SVF.

## B  Compare with other methods.

To clear the doubts of dataset, we use the unprocessed training set to make a fair comparison with other SOTA methods, as show in Table5. It can be seen that baseline with SVF achieves best performance on both Pascal-$5^i$ 1-shot and 5-shot settings. The experimental results prove that the advantages of SVF will not disappear due to the introduction of the training trick. Meanwhile, the experimental results prove that finetuning backbone is not only feasible in FSS, but also brings positive results to FSS models.

Table 5: Compare with SOTA on Pascal-$5^i$(8) in terms of mIoU for 1-shot and 5-shot segmentation.

| Method | backbone | 1-shot | | | | | 5-shot | | | | |
|---|---|---|---|---|---|---|---|---|---|---|---|
| | | Fold-0 | Fold-1 | Fold-2 | Fold-3 | Mean | Fold-0 | Fold-1 | Fold-2 | Fold-3 | Mean |
| PANet (10) | ResNet50 | 44.00 | 57.50 | 50.80 | 44.00 | 49.10 | 55.30 | 67.20 | 61.30 | 53.20 | 59.30 |
| CANet (14) | | 52.50 | 65.90 | 51.30 | 51.90 | 55.40 | 55.50 | 67.80 | 51.90 | 53.20 | 57.10 |
| PGNet (13) | | 56.00 | 66.90 | 50.60 | 50.40 | 56.00 | 57.70 | 68.70 | 52.90 | 54.60 | 58.50 |
| RPMM (11) | | 55.20 | 66.90 | 52.60 | 50.70 | 56.30 | 56.30 | 67.30 | 54.50 | 51.00 | 57.30 |
| PPNet (4) | | 47.80 | 58.80 | 53.80 | 45.60 | 51.50 | 58.40 | 67.80 | 64.90 | 56.70 | 62.00 |
| CWT (5) | | 56.30 | 62.00 | 59.90 | 47.20 | 56.40 | 61.30 | 68.50 | 68.50 | 56.60 | 63.70 |
| PFENet (9) | | 61.70 | 69.50 | 55.40 | 56.30 | 60.80 | 63.10 | 70.70 | 55.80 | 57.90 | 61.90 |
| CyCTR (15) | | 67.80 | 72.80 | 58.00 | 58.00 | 64.20 | 71.10 | 73.20 | 60.50 | 57.50 | 65.60 |
| baseline | | 66.36 | 69.22 | 57.64 | 58.73 | 62.99 | 70.75 | 72.92 | 58.86 | 65.56 | 67.02 |
| baseline + SVF | | 66.88 | 70.84 | 62.33 | 60.63 | **65.17** | 71.49 | 74.04 | 59.38 | 67.43 | **68.09** |

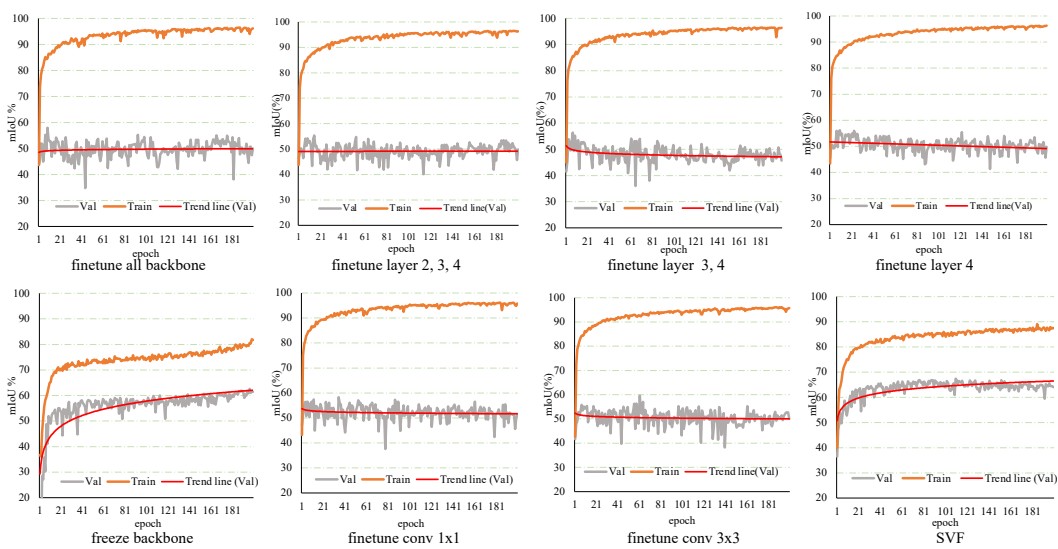

Figure 1: The mIoU curve of baseline with different finetune strategies on Pascal-5$^i$ Fold-0.

Table 6: Ablation study of BN on Pascal-5$^i$ under 1-shot setting. ✓represents fine-tuning this feature space. The best mean results are show in **bold**.

| Method | BN | scale | Fold-0 | Fold-1 | Fold-2 | Fold-3 | Mean |
|---|---|---|---|---|---|---|---|
| | | | 65.60 | 70.28 | 64.12 | 60.27 | 65.07 |
| | ✓ | | 61.93 | 70.67 | 62.02 | 57.86 | 63.12$_{(-1.95)}$ |
| baseline | ✓ | ✓ | 63.46 | 70.66 | 64.93 | 57.75 | 64.20$_{(-0.87)}$ |
| | | ✓ | 67.42 | 71.57 | 67.99 | 61.57 | **67.14**$_{(+2.07)}$ |

## C   Detailed Ablation Study

**Different finetune strategy:** In Figure 1, we visualize the mIoU curve of different fine-tuning strategies. It can be seen that both layer-based and convolution-based fine-tuning methods bring over-fitting problems. This result shows that traditional fine-tuning methods are not suitable for few-shot segmentation tasks. Directly fine-tuning the parameters of backbone in few-shot learning affects the robustness of FSS models. Therefore, we propose a novel fine-tuning strategy, namely SVF. It decompose pre-trained parameters into three successive matrices via the Singular Value Decomposition (SVD). Then, It only fine-tunes the singular value matrices during the training phase. The experimental results show that SVF can effectively avoid over-fitting while bringing positive results to FSS model.

**Sigular value subspace:** In Figure 2, we visualize the changes of initial Top-30 largest singular values of all $3 \times 3$ convolutional in layer 3 after SVF. The experimental results show that the change of last 3x3 convolution is the most obvious, and the change of singular value gradually moderates as the network becomes shallower. To verify the above point, we visualize the singular value change map of all 3x3 convolutions of layer 2 in Figure 3. The variation of singular values in layer2 is more gradual. Furthermore we visualize the singular value changes from the $1 \times 1$ convolution of layer 3 and layer 2 in Figure 4 and Figure 5. where the $1 \times 1$ convolution is the last $1 \times 1$ convolution of each block in ResNet. This result is the same trend as $3 \times 3$ convolution. It shown that the information concerned by deep convolutions in pre-train backbone is not conducive to few-shot segmentation tasks. SVF improves the expressiveness of FSS model by focusing on adjusting distribution of singular value subspace in the deep convolution. Meanwhile, It proves that semantic cues in deep convolutions have the greatest impact on few-shot segmentation. In addition, Figure 6 shows the variation of all singular values. It can be easy seen that the change of singular values afterward tends to 0. Therefore, the change of top-30 singular values can describe the change of all singular values.

In Table 6, Table 7, Table 8, Table 9 and Tbale 10, we give more detail ablation study results. It contains the results for each flod in different ablation study.

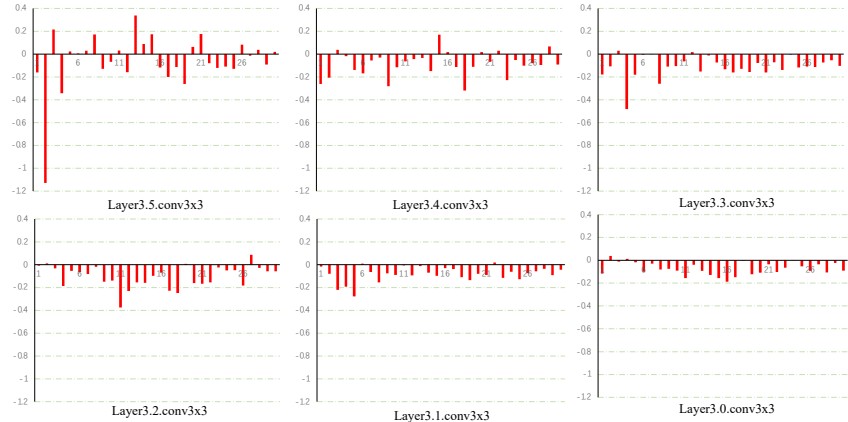

Figure 2: Statistics chart about the changes of initial Top-30 largest singular values of the $3 \times 3$ convolutional in layer3 after SVF.

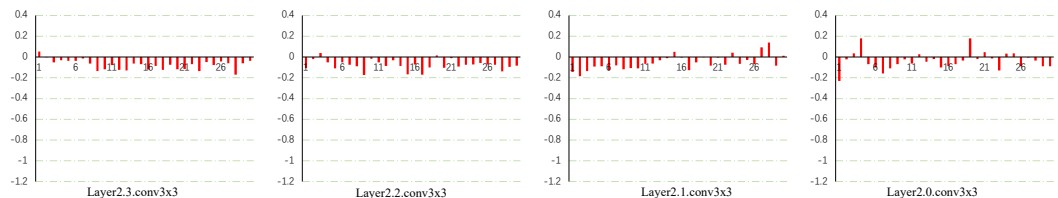

Figure 3: Statistics chart about the changes of initial Top-30 largest singular values of the $3 \times 3$ convolutional in layer2 after SVF.

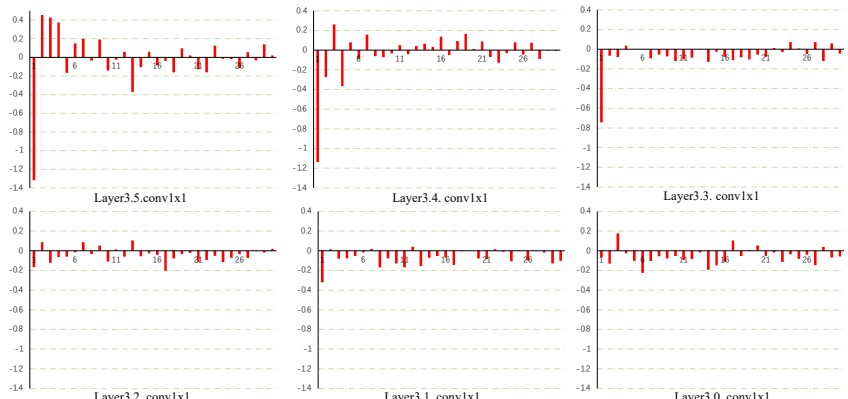

Figure 4: Statistics chart about the changes of initial Top-30 largest singular values of the $1 \times 1$ convolutional in layer3 after SVF.

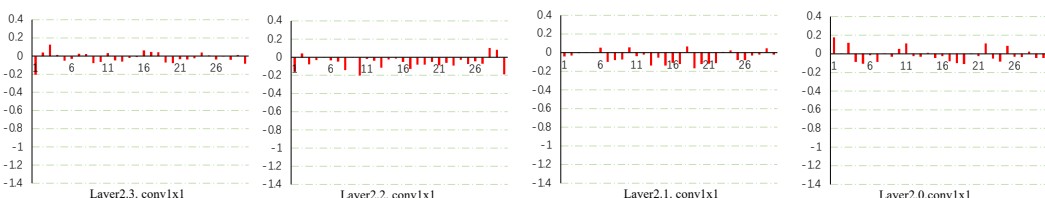

Figure 5: Statistics chart about the changes of initial Top-30 largest singular values of the $1 \times 1$ convolutional in layer2 after SVF.

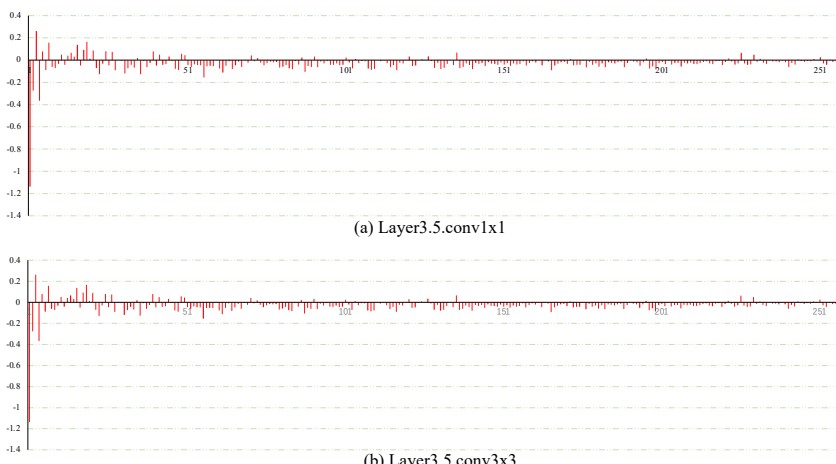

(a) Layer3.5.conv1x1

(b) Layer3.5.conv3x3

Figure 6: Statistics chart about the changes of all singular values of the last $3 \times 3$ and $1 \times 1$ convolutional in layer3 after SVF.

Table 7: Comparative experiment with fine-tuning different layer of backbone on Pascal-$5^i$.

| Method | layer | Fold-0 | Fold-1 | Fold-2 | Fold-3 | Mean |
|---|---|---|---|---|---|---|
| baseline | - | 65.60 | 70.28 | 64.12 | 60.27 | 65.07 |
| +fully fine-tune | 1, 2, 3, 4 | 57.97 | 70.51 | 61.33 | 53.80 | $60.90_{(-4.17)}$ |
| + part fine-tune | 2, 3, 4 | 55.34 | 71.16 | 62.72 | 55.38 | $61.15_{(-3.92)}$ |
|  | 3, 4 | 56.85 | 71.44 | 61.72 | 54.32 | $61.08_{(-3.99)}$ |
|  | 4 | 56.19 | 70.63 | 59.98 | 55.50 | $60.58_{(-4.49)}$ |
| +SVF | 2, 3, 4 | **67.42** | **71.57** | **67.99** | **61.57** | $\mathbf{67.14}_{(+2.07)}$ |

Table 8: Comparative experiment with fine-tuning different convolutional layer of backbone on Pascal-$5^i$.

| Method | layer | $3 \times 3$ | $1 \times 1$ | Fold-0 | Fold-1 | Fold-2 | Fold-3 | Mean |
|---|---|---|---|---|---|---|---|---|
| baseline | - | - | - | 65.60 | 70.28 | 64.12 | 60.27 | 65.07 |
| +part fine-tune | 2, 3, 4 | ✓ | ✓ | 55.34 | 71.16 | 62.72 | 55.38 | $61.15_{(-3.92)}$ |
|  | 2, 3, 4 | ✓ |  | 59.57 | 69.96 | 61.74 | 56.16 | $61.86_{(-3.21)}$ |
|  | 2, 3, 4 |  | ✓ | 58.30 | 70.50 | 62.04 | 55.63 | $61.62_{(-3.45)}$ |
| +SVF | 2, 3, 4 | - | - | 67.42 | 71.57 | 67.99 | 61.57 | $67.14_{(+2.07)}$ |

Table 9: Ablation study of SVF fine-tuning different subspace on Pascal-$5^i$.

| Method | U | S | V | Fold-0 | Fold-1 | Fold-2 | Fold-3 | Mean |
|---|---|---|---|---|---|---|---|---|
| baseline | ✓ |  |  | 58.14 | 70.06 | 60.91 | 55.24 | 61.09 |
|  |  | ✓ |  | 67.42 | 71.57 | 67.99 | 61.57 | 67.14 |
|  |  |  | ✓ | 53.87 | 70.63 | 63.65 | 55.36 | 60.88 |
|  | ✓ | ✓ |  | 57.54 | 70.19 | 62.12 | 56.41 | 61.57 |
|  |  | ✓ | ✓ | 53.30 | 71.21 | 62.24 | 54.92 | 60.42 |
|  | ✓ |  | ✓ | 53.81 | 70.75 | 61.92 | 53.60 | 60.02 |
|  | ✓ | ✓ | ✓ | 56.64 | 70.47 | 63.48 | 54.36 | 61.24 |

Table 10: Ablation study of SVF fine-tuning different layer on Pascal-$5^i$.

| Method | layer | Fold-0 | Fold-1 | Fold-2 | Fold-3 | Mean |
|---|---|---|---|---|---|---|
| baseline + SVF | 4 | 68.28 | 71.04 | 65.59 | 59.91 | 66.21 |
|  | 3, 4 | 67.21 | 71.88 | 68.12 | 61.57 | 67.20 |
|  | 2, 3, 4 | 67.42 | 71.57 | 67.99 | 61.57 | 67.14 |
|  | 1, 2, 3, 4 | 67.06 | 71.69 | 67.77 | 61.94 | 67.12 |

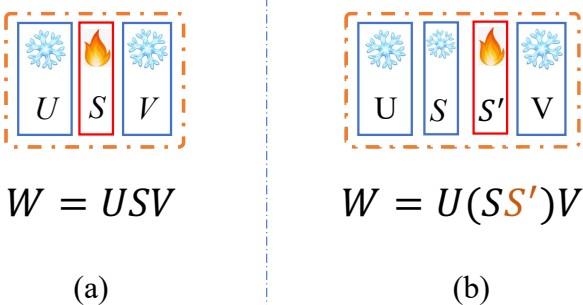

$$W = USV \qquad W = U(SS')V$$

(a) (b)

Figure 7: Different implementations of SVF.

Table 11: Comparing with only fine-tuning BN on Pascal-$5^i$.

| Method | Backbone | Fine-tuning Method | Fold-0 | Fold-1 | Fold-2 | Fold-3 | Mean |
|---|---|---|---|---|---|---|---|
| baseline | ResNet-50 | Freeze Backbone | 65.60 | 70.28 | 64.12 | 60.27 | 65.07 |
| | | Fine-tuning BN scale (weight) | 62.28 | 68.66 | 61.19 | 58.18 | 62.58 |
| | | Fine-tuning BN shift (bias) | 61.62 | 70.10 | 64.80 | 55.19 | 62.93 |
| | | Fine-tuning BN (weight+bias) | 61.93 | 70.67 | 62.02 | 57.86 | 63.12 |
| | | SVF | 67.42 | 71.57 | 67.99 | 61.57 | 67.14 |

# D  Discussion

## D.1  Discussion on other SVD

In this section, we discuss the differences between other SVD-based methods (1; 7) and SVF. Both SVB (1) and *Hanie* (7) constrain the distribution of the singular values $s$ where SVB (1) forces the singular value around 1 and *Hanie* (7) clamps the large singular values into a constant, hence serving as a regularization term. We did not pose an extra constraint on $s$, instead, encouraged the fully trainable singular values. As illustrated in SVB's Figure 1, the singular values of well-trained weights are widely spread around [0,2]. The strong regularization proposed in SVB (1) and *Hanie* (7) should damage the performance of pre-trained networks. Therefore, they turn to training from scratch, which is infeasible in the circumstance of few-shot segmentation. Our method coupled with pre-trained parameters can further exploit the capacity of the backbone, leading to superior results.

## D.2  Discussion on different implementation

In this section, we provide a discussion on our SVF. The main idea of SVF is learning to change singular values in the backbone weights. It has different implementations. We show two possible ways to achieve SVF in Figure 7: (i) treat the single value matrix $S$ as trainable parameters directly; (ii) freeze the original singular value matrix $S$ and introduce another trainable singular value matrix $S'$ (we use exponential function *exp* to keep it positive and initialize it with zeros), where the final singular value matrix is a product of $S$ (frozen) and $S'$ (trainable). In the second implementation, SVF keeps the backbone frozen (as all its weights are frozen) while introducing a small part of extra trainable parameters. It shares similarities with the recently proposed Visual Prompt Tuning (VPT) (2). The difference between VPT and SVF is that VPT introduces the trainable parameters in the input space while SVF introduces them in the singular value space. Although SVF and VPT freeze the original backbone, they can produce optimization on the feature maps of the backbone. This property enables SVF to perform better in few-shot segmentation (FSS) and is the essential

Table 12: introduce a new small part of parameters S' to verify the importance of singular values on Pascal-$5^i$.

| Method | Backbone | Expression of weight | Fine-tune param | Fold-0 | Fold-1 | Fold-2 | Fold-3 | Mean |
|---|---|---|---|---|---|---|---|---|
| baseline | ResNet-50 | W | - | 65.60 | 70.28 | 64.12 | 60.27 | 65.07 |
| | | S'W | S' | 60.96 | 71.99 | 62.54 | 58.58 | 63.52 |
| | | WS' | S' | 62.82 | 71.69 | 62.84 | 61.13 | 64.62 |
| | | $USV^T$ | S | 67.42 | 71.57 | 67.99 | 61.57 | **67.14** |

Table 13: Compare with different implementations of SVF on Pascal-$5^i$ 1-shot.

| Method | Backbone | Expression of weight | Fine-tune param | Fold-0 | Fold-1 | Fold-2 | Fold-3 | Mean |
|---|---|---|---|---|---|---|---|---|
| baseline | ResNet-50 | W | - | 65.60 | 70.28 | 64.12 | 60.27 | 65.07 |
| | | $USV^T$ | S | 67.42 | 71.57 | 67.99 | 61.57 | **67.14** |
| | | $USS'V^T$ | S' | 67.16 | 71.58 | 68.59 | 61.08 | 67.10 |
| | | $USS'V^T$ | S + S' | 66.42 | 71.73 | 67.23 | 61.12 | 66.63 |

Table 14: Compare with other SVD-based methods on Pascal-$5^i$ 1-shot.

| Method | Backbone | Expression of weight | Fine-tune param | Fold-0 | Fold-1 | Fold-2 | Fold-3 | Mean |
|---|---|---|---|---|---|---|---|---|
| baseline | ResNet-50 | W | - | 65.60 | 70.28 | 64.12 | 60.27 | 65.07 |
| | | $USV^T$ | S | 67.42 | 71.57 | 67.99 | 61.57 | **67.14** |
| | | S'W | S' | 60.96 | 71.99 | 62.54 | 58.58 | 63.52 |
| | | RS'R'W | S' | 32.91 | 51.93 | 51.00 | 37.60 | 43.36 |

difference from the properties in previous SSF methods with frozen backbone (they do not change the feature maps of the backbone).

### D.3 Discussion on success of SVF

In this section, we discuss the truly responsible for the success of SVF from three question. First, Does fine-tune another small part of parameters in the backbone work? We conduct experiments on Pascal-$5^i$ with the 1-shot setting. We compare our SVF with methods that only fine-tune the parameters in the BN layers. The results in Table 11 show that only fine-tuning the parameters in BN layers does not bring over-fitting in few-shot segmentation methods, but they perform worse than the conventional paradigm (freezing backbone). While our SVF outperform other methods by large margins.

Second, Is it really necessary to fine-tune the singular values? What if we introduce a new small part of parameters S', which is not in the singular value space, and only fine-tune the S'? To answer this question, we conduction two experiments, where the weight becomes S'W or WS', and only fine-tune the introduced small part of parameters S'. The results in Table 12 are consistence with Table 11. Both of them can avoid over-fitting but show slightly worse performance than the freezing backbone baseline. The above experimental results suggest that fine-tuning a small part of parameters is a good way to avoid over-fitting when fine-tuning the backbone in few-shot segmentation. But it is non-trivial to find such a small part of parameters that can bring considerable improvements.

Third, What causes the differences between SVF and WS' or S'W? In this question, we try to provide our understanding of what causes the superior performances of SVF over WS' and S'W. We conjecture that this may be related to the context that S or S' can access when fine-tuning the parameters. Assume that W has the shape of $[M, N]$. S and S' are diagonal matrices. S has the shape of [Rank, Rank], and S' has the shape of $[M, M]$ or $[N, N]$. When optimizing the parameters, S' only has relations on dimension M or dimension N in a channel-wise manner, while S can connect all channels on both dimension M and dimension N, as S is in the singular value space. This differences can affect the received gradients when training S or S', which results in different performance. To give more evidences, we design more variants of SVF and provide their results in Table 13.

Finaly, To verify whether SVF depends crucially on the singular value space, or simply on the number of effective updated parameters. we design a experiment: let R be a random rotation matrix, and set U=R' and V=RW, where W is the original weight matrix for the given layer. The formulation of the weight becomes RS'R'W. Note that S' is initialized with an identity matrix as done in previous experiments. During the fine-tuning, we only train S' while keep others frozen in the backbone. We provide the results in Table 14. Random rotation formulation gives poor results. In fact, if we set R as an identity matrix (identity matrix is a rotation matrix), RS'R'W = S'W. As shown in the table, S'W is much better than random RS'R'W. It seems that the selection of the rotation matrix R is critical to the final segmentation performance. Meanwhile, If we consider RS'R' (it is a diagonal matrix in the initialization stage) as a whole, RS'R is only related to one dimension of the weight W. Thus for the middle matrix S', it is also channel-aligned with respect to weight W.

In addition, if R is random initialized, we can not guarantee that RS'R' is a diagonal matrix when updating S' during training (we verify this phenomenon with the saved checkpoints when we finish the

training). Note that the weight W is the one from the pre-trained backbone, which contains semantic clues or learned knowledge. The non-diagonal matrix RS'R' may bring unexpected transformation to the pre-trained weight W, leading to poor results.