# OpenReview forum: "Singular Value Fine-tuning: Few-shot Segmentation requires Few-parameters Fine-tuning"
_NeurIPS.cc/2022/Conference — NeurIPS 2022 Accept_

### Official Review · Reviewer_jAEv · 2022-06-26

**Rating:** 8
**Confidence:** 4
**Soundness:** 4 excellent
**Presentation:** 4 excellent
**Contribution:** 4 excellent

**Summary:**

In this paper, the authors propose a novel SVF to changes the standard paradigm of freeze backbone in few-shot segmentation.The results show that only fine-tune the parameters of the singular value subspace (S) and freeze other subspaces can not only effectively avoid the overfitting problem, but also significantly improve the performance of FSS model. Specifically, the visualization results prove that some semantic cues with high weight in the pretrained weight are not conducive to downstream tasks, thus SVF adjusts the weights of different semantic cues by fine-tuning the parameters of subspace S. They further confirm the effectiveness of their proposed method by running experiments on real datasets and applications and comparing them to other topologies and methods. Moreover, they further confirm the effectiveness and superiority of SVF by comparing them to other representative methods on two common datasets.

**Questions:**

- In section 3.3 and appendix, the author details two perspectives of SVF. The two perspectives are implemented differently. The paper does not analyze the impact of SVF from different perspectives on the FSS model.

- As an important part of the backbone, the bias has a huge impact on the model during the finetuning process. The authors lack an analysis of bias among different fine-tuning methods.

- In section 4.4 and apendix, the author only shows the change of the top-30 singular values of different convolutions, and the change results of all the singular values are missing.


**Ethics Review Area:**

["I don’t know"]

**Limitations:**

This paper does not reflect any potential negative societal impact.

**Strengths And Weaknesses:**

- This paper is overall writing is clear and easy to follow. This is a really interesting paper. This paper not only analyzes the reason why the freezing backbone in FSS has become the traditional paradigm, but also proposes a novel fine-tuning method to try to break the traditional paradigm. The novel method SVF design is simple yet effective as proven by extensive experiments and analysis. Meanwhile, this paper analyzes the impact of hidden tricks in FSS on model performance to facilitate fair comparisons in future work. The core of SVF is a novel method of fine-tuning backbone, thus SVF has universality and can be easily used for all FSS models. As a result, this could serve as a new paradigm for few-shot semantic segmentation task. I believe that this paper will be a valuable contribution to the field, and I strongly recommend acceptance.

- The experimental results demonstrate the effectiveness of SVF, but will increase the training cost by a small amount compared to freeze backbone.

---

> ### Author Response · Authors · 2022-08-02
> **Response to reviewer jAEv**
>
> Thanks a lot for your time and feedback. Below are address all raised concerns of the paper.
>
> ---
>
> **Q1: The paper does not analyze the impact of SVF from different perspectives**
>
> **A1:** Thanks for pointing it out. The two perspectives of SVF are theoretically equivalent, and the purpose of fine-tuning S and S' is to change distribution of the singular values space. S' of the other implementations of SVF is a learnable parameter initialized to 1, and its size is the same as S. From a theoretical point of view S = SS', therefore SS' constitutes a new S. Below we compare the performance of the models under two perspectives.
>
> |Method	|Backbone	|init	|Fine-tune param|Fold-0|	Fold-1|	Fold-2|	Fold-3|	Mean|
> | ------------ | ------------ | ------------ | ------------ | ------------ | ------------ | ------------ | ------------ | ------------ |
> | baseline + SVF|ResNet-50|-	|S|67.42	|71.57	|67.99	|61.57	|67.14 |
> | baseline + SVF' |ResNet-50   |  1 |S'| 67.16|	71.58|	68.59|	61.08|	67.10|
> | baseline + SVF' | ResNet-50  |  0 with exp|S'| 67.50|	72.35|	67.70|	61.66|	**67.30**  |
>
> where SVF' represents other implementations of SVF. The experimental results show that when S' initialized to 1, the performance of SVF under both views is consistent. SVF performs better when initialized to 0 with exp. The exp adds nonlinear factors to SVF, which further improves the expressiveness of SVF. It shows that SVF has the possibility of further improvement.
>
> ---
>
> **Q2: Compare with bias tuning**
>
> **A2:** Thanks for pointing it out. In the ResNet backbone, the convolution layers do not contain bias term. The bias terms that can be used for tuning is the ones in BN layers. Below we supplement the test results of bias in the fine-tune the bias terms in all BN layers.
>
> |  Method|Backbone| Finetuning method|	Fold-0|	Fold-1|	Fold-2|	Fold-3|	Mean  |
> | ------------ | ------------ | ------------ | ------------ | ------------ | ------------ | ------------ | ------------ |
> | baseline  |  ResNet-50 |Freeze Backbone |  65.60  | 70.28  | 64.12  | 60.27  | 65.07  |
> | baseline  |  ResNet-50 | SVF  | 67.42|71.57|	67.99|	61.57|	**67.14** |
> |  baseline |  ResNet-50 | Bias-Tuning |61.62|	70.10|	64.80|	55.19|	62.93 |
>
> The experimental results show that bias tuning does not achieve better results than freeze backbone. The BN layer does not contain semantic information, and the convolution layer does not contain bias. Therefore, the bias-tuning cannot have a positive impact on the few-shot segmentation model
>
> ---
>
> **Q3: The change of all singular values of different convolutions**
>
> **A3:** Thanks for pointing it out. We add the change of all singular values for different convolutions to revised appendix. The changes of singular values reveal the importance of different semantic cues in the backbone to downstream tasks. We find that the singular value change after TOP-30 tends to 0. Therefore, we believe that TOP-30 can describe the variation of all singular values.

---

> > ### Comment · Reviewer_jAEv · 2022-08-08
> > **Thanks for the authors’feedback**
> >
> > The response by the authors addressed my concerns well, also consider the comments from other reviewers, I will keep my previous rating of this paper and suggest to **strong accept** this paper.

---

> ### Author Response · Authors · 2022-08-06
> **Follow-up questions:**
>
> We appreciate your valuable comments. We were wondering if our responses have addressed your concerns. Please let us know if you have additional questions. Thank you!

---

### Official Review · Reviewer_LFht · 2022-07-07

**Rating:** 7
**Confidence:** 5
**Soundness:** 4 excellent
**Presentation:** 4 excellent
**Contribution:** 4 excellent

**Summary:**

The authors have proposed a new training scheme for FSS frameworks by only adapting a few parameters of the ImageNet pre-trained backbone to the segmentation task. The core idea is to adjust the singular values of the pre-trained kernel weights of the backbone, bringing considerable improvements to representative FSS methods (PFENet and BAM). Extensive experiments have shown the effectiveness of the proposed method.

**Questions:**

This paper does present a good method for boosting existing few-shot segmentation methods whose backbone parameters are fixed, and considerable improvements have been achieved. However, my biggest concern is still about the training setting.

The authors of submission 1788 follow BAM to remove all training images that contain novel classes, for the purposes of avoiding information leakage, but all previous methods in the community follow Shaban et al[1] to keep those images during the training phase by setting the labels for the novel classes as background, which explains why PFENet without dagger in Table 1 is much worse than the one with the dagger.

More specifically, in section 3 of the paper [1], the authors of [1] wrote ''In this problem, unlike image classification, examples from L_test might appear in training images. This is handled naturally when an annotator unaware of some object class, labels it as background''.

Therefore, the reviewer thinks that BAM actually has introduced an unfair comparison in their paper, and it would be better if the authors of submission 1788 could clearly present the results of Baseline + SVF, PFENet + SVF, and BAM + SVF in Table 1 and Table 2, by keeping the novel categories but setting them as the background during the base training phase, for a fair comparison with previous methods that adopt the same setting as described in [1].

In Table 2 of the appendix, what causes the success on Fold-2 when the novel classes are removed from the training set? If the novel classes are included as [1] and set as the background, whether the proposed method will be negatively affected?

Minor issues:
1. Are the PFENet and BAM without dagger shown in table 3 trained with novel classes whose labels are set to the background?

2. Visualizations in Figure 5 and Figure 6 are confusing and they could be improved by indicating what the target classes are. For example, in the last examples of Figure 5 and Figrue 6, the 1x1 weights focus on the person more than boat, but contrarily, the 3x3 weights are more curious about the boat, which contradicts the problem setting that only one target class exists in each evaluation episode.


References
[1]  One-Shot Learning for Semantic Segmentation. BMVC 2017



**Limitations:**

As described in the Questions section, there might be some unfair comparisons with previous methods, and some critical aspects are not clear enough. The authors are encouraged to show additional results to support their claims.

[updated after rebuttal] My initial concerns regarding the comparison have been well-addressed in the revision. Thus my final rating is ACCEPT (increased from 4 -> 7).

**Strengths And Weaknesses:**

Strength:

+ A good extension of the current training scheme for FSS frameworks.

+ Decent performance gain has been brought to representative FSS methods (PFENet and BAM).

+ Clear motivation and good presentation of the method and discussion.

+ The overall submission is well-prepared with comprehensive appendix.

Weakness:

- Unfair comparisons with the previously proposed methods (please see Questions section for details).

- Visualizations in Figure 5 and Figure 6 are confusing.

If the above weaknesses could be well addressed, I would like to give a higher rating.

---

> ### Author Response · Authors · 2022-08-02
> **Response to reviewer LFht**
>
> Thank you for your valuable feedback! Below are address all raised concerns of the paper.
>
> ---
>
> **Q1: Unfair comparisons with the previously proposed methods in Table 1 and Table 2.**
>
> **A1:** The purpose of Table 1 and Table 2 is to verify the effectiveness of SVF under the same training setting, and to verify the universality of SVF on different methods. Our purpose in adding PFENet (without dagger) results is to hope that researchers will notice the impact of dataset trick on FSS model performance. And, we perform the detailed analysis of dataset trick in appendix.
>
> We agree with the reviewer about the dataset trick affects performance of FSS model. Therefore, we supplement the analysis of some unfair training tricks in few-shot segmentation on appendix. And we provide fair comparison results with previously proposed methods in appendix Table 4. All methods in Table 4 adopt the same setting as described in [A]. Below we provide the results of Baseline, PFENet, and BAM with or without SVF in Pascal-5$^i$ 1-shot. Following [A], the dataset used in this experiment did not remove images containing the novel classes from the training set.
>
> |  Method |Backbone   | Training Trick  | Fold-0  |  Fold-1 | Fold-2  | Fold-3  |  Mean |
> | ------------ | ------------ | ------------ | ------------ | ------------ | ------------ | ------------ | ------------ |
> | baseline  |  ResNet-50 |w/o   | 66.36|	69.22|	57.64|	58.73|	62.99|
> | baseline + SVF  |  ResNet-50 | w/o  |66.88	|70.84|	62.33|	60.63|	**65.17**|
> | PFENet  |  ResNet-50 | w/o  |67.06	|71.61	|55.21	|59.46	|63.34  |
> | PFENet + SVF  |  ResNet-50 |  w/o |68.31	|71.99	|56.25	|61.82	|**64.59**|
> | BAM  |  ResNet-50 |  w/o | 68.37	|72.05	|57.55	|60.38	|64.59 |
> | BAM + SVF |  ResNet-50 |  w/o | 68.17	|72.86	|57.77	|62.04	|**65.21**|
>
> Experimental results show that retaining novel categories in the base training stage and setting them as backgrounds does not negatively affect SVF. It also shows that whether or not the dataset tricks is used does not affect the effectiveness of SVF. The purpose of our detailed discussion of training trick is to promote the development of community health.
>
> ---
> **Q2: what causes the success on Fold-2 when the novel classes are removed from the training set?**
>
> **A2:** Below we count the number of images in each fold before and after using the dataset trick.
>
> | Pascal 5$^i$  | Fold-0  |  Fold-1  |   Fold-2 | Fold-3   |
> | ------------ | ------------ | ------------ | ------------ | ------------ |
> | w/o remove novel classes  | 4760	|4588	|4097	|5108 |
> | remove novel classes	|4208	|3726	|2752	|4510 |
> | reduction rate	|11.6%	|18.8%	|**32.8%**	|11.7% |
>
> The statistical results show that the number of images containing novel classes in Fold-2 training set is 2-3 times that of other folds. We guess that the removed images negatively affect the results of Fold-2. Therefore, the performance improvement of Fold-2 is most obvious when removing images containing novel classes in training set.
>
> ---
>
> **Q3: Visualizations in Figure 5 and Figure 6 are confusing.**
>
> **A3:** In Figures 5 and 6 we use images from the Fold-1 training set. We guess that the semantic cues with the largest singular value growth are conducive to Few-shot segmentation, therefore the base class area will be displayed in the visualization results. Below, we show base classes of each fold on Pascal-5$^i$. It can be seen that both 'boat' and 'person' are in the base classes of Fold-1. Therefore, the weight after finetuning focus on not only the 'person', but also the 'boat'.
>
> |  Pascal-5$^i$ | base classes  |
> | ------------ | ------------ |
> | Fold-0  | bus, car, cat, chair, cow, diningtable, dog, horse, motorbike, person,potted plant, sheep, sofa, train, tv/monitor |
> | **Fold-1**  | aeroplane, bicycle, bird, **boat**, bottle,diningtable, dog, horse, motorbike, **person**, potted plant, sheep, sofa, train, tv/monitor   |
> | Fold-2  | aeroplane, bicycle, bird, boat, bottle, bus, car, cat, chair, cow, potted plant, sheep, sofa, train, tv/monitor |
> | Fold-3  | aeroplane, bicycle, bird, boat, bottle, bus, car, cat, chair, cow, diningtable, dog, horse, motorbike, person |
>
> ---
>
> [A] Shaban, Amirreza, Shray Bansal, Zhen Liu, Irfan Essa, and Byron Boots: One-Shot Learning for Semantic Segmentation. BMVC. 2017.

---

> > ### Comment · Reviewer_LFht · 2022-08-03
> > **Please add the experiments to the main body**
> >
> > The reviewer appreciates the authors' responses.
> >
> > Still, the reviewer wants to ask the authors whether they can **put the results in the rebuttal Q1/A1 to the main body of the submission, instead of the appendix**. It would be much better if these results are clearly added to Table 1 and Table 2 in the main submission. Please submit a new revision if it is allowed, and the reviewer will give a quick review again.
> >
> > The reviewer encourages this necessary action and believes it may help broader readers be aware of the effects brought by **the extremely unfair training setting** introduced by BAM, which may help the community grow better by retaining a fair performance comparison that could better tell the effectiveness of the proposed SVF.

---

> > > ### Author Response · Authors · 2022-08-03
> > > **We revise the paper and add fair comparisons to the main body**
> > >
> > > We agree with the reviewer that fair comparisons should be conducted when proving the effectiveness of the proposed method. Following the reviewer's suggestion, we upload a new revision, where we discuss the dataset trick brought by BAM in page 6 Section 4.1 (marked in red). We add the results in Q1/A1 to Table 1 in the main body page 5 (marked in red). For the 5-shot setting, we are running new experiments to get the results. We will update them to Table 1 in the final version.
> > > Please let us know if the reviewer has further suggestions about the comparisons.

---

> > > > ### Comment · Reviewer_LFht · 2022-08-04
> > > > **My concerns are well addressed.**
> > > >
> > > > The authors' responses have addressed my concerns. My final rating is **ACCEPT**, and the initial review has been updated.
> > > >
> > > > In general, this paper presents a new direction for promoting the research of FSS, and the method itself is novel to the community. In particular, the proposed method is applicable to various models without structural constraints. After rebuttal, my initial concerns regarding the comparison have been well addressed by the authors in that they have supplemented additional experimental results and in-depth discussions.
> > > >
> > > > Also, I have read Reviewer xj2U's comments, and I encourage the authors to put the experimental comparisons with Adaptor and Bias tunning to the appendix for better completeness.
> > > >
> > > > To this end, given the convincing new experimental results and discussions, **I vote and argue for accepting this paper** and hope the authors could **open-source the related implementations**.

---

> > > > > ### Author Response · Authors · 2022-08-06
> > > > > **Thanks for your positive feedback.**
> > > > >
> > > > > Thanks for your positive feedback. We think the suggested changes and additions made here have greatly improved the work.
> > > > >
> > > > > Following the suggestions, We add the 5-shot experimental results in the main body page 5 (marked in red). And the experimental comparisons with Adaptor and Bias tuning are added in revision appendix section B.2.
> > > > >
> > > > > In addition, we will open all the source code of SVF later.

---

### Official Review · Reviewer_8FCE · 2022-07-11

**Rating:** 7
**Confidence:** 5
**Soundness:** 3 good
**Presentation:** 3 good
**Contribution:** 3 good

**Summary:**

Authors re-examine the idea of fine-tuning the backbone feature extractor during few-shot semantic segmentation, showing that overfitting can be avoided by limiting updates to a small set of parameters. Specifically, authors decompose convolution layers using SVD, and fine tune only the singular values S. Results indicate that this approach increases performance relative to using a frozen backbone, while fine-tuning other groups of parameters consistently decreases performance.

**Questions:**

What is truly responsible for the success of SVF?

I propose the following experiment to start: let R be a random rotation matrix, and set U=R’ and V=RW, where W is the original weight matrix for the given layer. Then attempt SVF fine-tuning. This will show whether SVF depends crucially on the singular value space, or simply on the number of effective updated parameters. Fine tuning only the BN scale terms might also be worth a try, for a true apples-to-apples comparison – it could be that the bias term is uniquely destructive.

Alternatively, if authors could elaborate on what makes the singular value space so unreasonably amenable to fine-tuning, this may become clearer. Perhaps I am simply failing to understand the intended argument.

SMALL COMMENTS AND TYPOS:

Pg3 line 100: while does not  while it does not

Pg4 line 103: one need  one needs

Pg4 line 132: parameters, also  parameters, and also

Pg5 line 177: all classes all classes  all classes

Pg5 lines 178-179: are used … are used  used … used

Pg7 line 214: shows that the  shows the

Pg7 lines 227-228: you seem to have a redundant sentence here (Finally… Finally)

Pg8 line 251: without destroy  without destroying


**Limitations:**

Discussion of limitations is fair. Societal impacts are not discussed, though do not extend beyond those of few-shot learning in general.

**Strengths And Weaknesses:**

STRENGTHS:

Motivation is clear, and approach is sensible, straightforward, and highly applicable. Results are convincing and a thorough analysis is provided. Paper is well organized (though with occasional typos and some awkward language, see below).

WEAKNESSES:

While the paper convincingly demonstrates that SVF is effective, and acts as expected, it does not adequately explain why. Authors imply that fine-tuning in the singular value space is uniquely non-destructive, but it is not intuitively obvious that this should be the case. Indeed, in Fig.4 many values
switch signs, indicating that nothing is stopping SVF from zeroing out large swaths of the output manifold in practice. Without knowing why the singular value space is so particularly effective, the contribution is limited, as constrained fine-tuning is already a widely known approach in the few-shot regime (see below) and is not novel in and of itself.

Along these lines, while the analysis is broad and involved, the comparisons are somewhat apples-to-oranges. Rather than singular values being special, it could instead be that SVF simply allows the authors to fine-tune a smaller number of parameters than their comparative baselines. For starters, all fine-tuning comparisons except for BN involve a far larger number of parameters, and so the observed overfitting is not surprising, and even the BN baseline involves twice as many parameters (scale _and_ shift). Additionally, it could be that SVF allows for a smaller number of _effective_ updates compared to BN. For example, if matrix U is highly axis-aligned, changes to S will shrink or vanish in the subsequent BN layer. If U=I, then S should not update at all.

Further analysis is required.

Less importantly: while the related work is quite broad, there exist similar approaches in the broader few-shot literature also based on fine-tuning a highly constrained subset of introduced parameters. These may be worth mentioning – e.g. LEO (Meta-Learning with Latent Embedding Optimization, ICLR2019), CNAPS (Fast and Flexible Multi-Task Classification Using Conditional Neural Adaptive Processes, NeurIPS2019), or possibly FiLM (FiLM: Visual Reasoning with a General Conditioning Layer, AAAI2018).

---

> ### Author Response · Authors · 2022-08-02
> **Response to reviewer 8FCE (1/2)**
>
> Thanks a lot for your time and feedback. We have to say that the reviewer asks valuable questions and provides thoughtful clues. We appreciate your inspiring reviews. And we are happy to address the concerns.
>
> ---
>
> The main question of the reviewer is: what is truly responsible for the success of SVF? According to the reviewer's comments and suggestions, we split it into three sub-parts:
>
> - Does fine-tune another small part of parameters in the backbone work? Comparing with only fine-tuning BN can be a good example.
> - Is it really necessary to fine-tune the singular values? What if we introduce a new small part of parameters S', which is not in the singular value space, and only fine-tune the S'? (We simplify the experiment posed by the reviewer to fine-tuning the S' in S'W or WS'. As R is a random rotation matrix, thus R'R=I. )
> - What causes the differences between SVF and WS' or S'W?
>
> We give the responses below.
>
> ---
> **Q1: Does fine-tune another small part of parameters in the backbone work? Comparing with only fine-tuning BN can be a good example.**
>
> **A1:** We conduct experiments on Pascal-5$^i$ with the 1-shot setting. We compare our SVF with methods that only fine-tune the parameters in the BN layers. The results below show that only fine-tuning the parameters in BN layers does not bring overfitting in few-shot segmentation methods, but they perform worse than the conventional paradigm (freezing backbone). While our SVF outperform other methods by large margins.
>
> |  Method |Backbone   | Fine-tuning Method  | Fold-0  |  Fold-1 | Fold-2  | Fold-3  |  Mean |
> | ------------ | ------------ | ------------ | ------------ | ------------ | ------------ | ------------ | ------------ |
> | baseline  |  ResNet-50 |Freeze Backbone   |  65.60  | 70.28  | 64.12  | 60.27  | 65.07  |
> | baseline  |  ResNet-50 | Fine-tuning BN scale (weight)  | 62.28  | 68.66  | 61.19  | 58.18  | 62.58  |
> | baseline  |  ResNet-50 | Fine-tuning BN shift (bias)  | 61.62 | 70.10  | 64.80  | 55.19  | 62.93  |
> | baseline  |  ResNet-50 |  Fine-tuning BN (weight+bias) | 61.93  | 70.67  | 62.02  | 57.86  | 63.12  |
> | baseline  |  ResNet-50 |SVF   |  67.42  | 71.57  | 67.99  | 61.57  | **67.14**  |
>
> ---
>
> **Q2: Is it really necessary to fine-tune the singular values? What if we introduce a new small part of parameters S', which is not in the singular value space, and only fine-tune the S'?**
>
> **A2:** To answer this question, we conduction two experiments, where the weight becomes S'W or WS', and only fine-tune the introduced small part of parameters S'. The results are consistence with the experiments in Q1. Both of them can avoid overfitting but show slightly worse performance than the freezing backbone baseline.
>
> |  Method | Backbone  | Expression of weight  | Fine-tune param  | Fold-0  | Fold-1  | Fold-2  | Fold-3  | Mean  |
> | ------------ | ------------ | ------------ | ------------ | ------------ | ------------ | ------------ | ------------ | ------------ |
> | baseline  | ResNet-50  | W  | -  | 65.60| 70.28| 64.12| 60.27| 65.07 |
> | baseline  | ResNet-50  | S'W  | S'  | 60.96 | 71.99 | 62.54 | 58.58 | 63.52  |
> | baseline  | ResNet-50  | WS'  | S' | 62.82  | 71.69  | 62.84  | 61.13  | 64.62 |
> | baseline  | ResNet-50  | USV$^T$  | S |  67.42  | 71.57  | 67.99  | 61.57  | **67.14**  |
>
> The above experimental results in Q1 and Q2 suggest that fine-tuning a small part of parameters is a good way to avoid overfitting when fine-tuning the backbone in few-shot segmentation. But **it is non-trivial to find such a small part of parameters that can bring considerable improvements**.

---

> > ### Author Response · Authors · 2022-08-02
> > **Response to reviewer 8FCE (2/2)**
> >
> > **Q3: What causes the differences between SVF and WS' or S'W?**
> >
> > **A3:** In this question, we try to provide our understanding of what causes the superior performances of SVF over WS' and S'W. We conjecture that this may be related to the context that S or S' can access when fine-tuning the parameters. Assume that W has the shape of [M, N]. S and S' are diagonal matrices. S has the shape of [Rank, Rank], and S' has the shape of [M, M] or [N, N]. When optimizing the parameters, S' only has relations on dimension M or dimension N in a channel-wise manner, while S can connect all channels on both dimension M and dimension N, as S is in the singular value space. This differences can affect the received gradients when training S or S', which results in different performance. To give more evidences, we design more variants of SVF and provide their results in the table below.
> >
> > | Mehod  | Backbone  |Expression of weight |Fine-tune param | Fold-0  |  Fold-1 | Fold-2  | Fold-3  |  Mean |
> > | ------------ | ------------ | ------------ | ------------ | ------------ | ------------ | ------------ | ------------ | ------------ |
> > | baseline  | ResNet-50  | W  | -  | 65.60| 70.28| 64.12| 60.27| 65.07 |
> > |  baseline |ResNet-50  |USV$^T$|S|  67.42  | 71.57  | 67.99  | 61.57  | 67.14  |
> > |  baseline | ResNet-50 |USS'V$^T$|S'|  67.16 | 71.58  | 68.59  | 61.08  | 67.10  |
> > |  baseline | ResNet-50 |USS'V$^T$| S + S'|  66.42  | 71.73  | 67.23  | 61.12  | 66.63  |
> >
> > We find that given S and S' are lie in the singular value space, all variants can outperform the freezing backbone baseline.
> >
> > ---
> >
> > **Q4: references and typos.**
> >
> > **A4:** Thanks for pointing it out. We add all the related literature in our revised version. And we fix the typos. The modification are illustrated with red color in the revised paper.
> >
> > ---

---

> > > ### Comment · Reviewer_8FCE · 2022-08-03
> > > **Thank you for the new results; review updated**
> > >
> > > Thank you for the in-depth response, the new results and analysis are greatly appreciated. I consider my main concern partially addressed at this point: authors have fairly convincingly demonstrated that tuning in the singular value space is crucial (and please do add these results to paper or supplementary!) but I’m still not sure what is responsible for this fact. Authors postulate that the US’V formulation outperforms the S’W and WS’ formulations because it is not channel-aligned, and thus better contextualized w.r.t. gradient updates. This could be the case, but this argument would also apply to the original review’s proposed random rotation formulation RS’R’W (with some abuse of notation in the apostrophe), which is not evaluated. The fact that SVF is not channel-aligned, and therefore outperforms channel-aligned alternatives, does not in and of itself explain why we should use the singular value space in particular. However, the new results do provide necessary clarity, so I have updated my review to Weak Accept.

---

> > > > ### Author Response · Authors · 2022-08-05
> > > > **Response to reviewer 8FCE**
> > > >
> > > > We thank the reviewer for providing detailed illustration about the random rotation matrix setting. We conduct a new experiment with a random initialized rotation matrix R (we use the scipy.stats.special_ortho_group function). The formulation of the weight becomes RS’R’W. Note that S’ is initialized with an identity matrix as done in previous experiments. During the fine-tuning, we only train S’ while keep others frozen in the backbone. We provide the results below. Random rotation formulation gives poor results.
> > > >
> > > > | Mehod  | Backbone  |Expression of weight |Fine-tune param | Fold-0  |  Fold-1 | Fold-2  | Fold-3  |  Mean |
> > > > | ------------ | ------------ | ------------ | ------------ | ------------ | ------------ | ------------ | ------------ | ------------ |
> > > > | baseline  | ResNet-50  | W  | -  | 65.60| 70.28| 64.12| 60.27| 65.07 |
> > > > |  baseline |ResNet-50  |USV$^T$|S|  67.42  | 71.57  | 67.99  | 61.57  | 67.14  |
> > > > | baseline  | ResNet-50  | S'W  | S'  | 60.96 | 71.99 | 62.54 | 58.58 | 63.52  |
> > > > |  baseline |ResNet-50  |RS'R'W|S'|  32.91  | 51.93  | 51.00  | 37.60  | 43.36  |
> > > >
> > > > We try to explain the results:
> > > >
> > > > 1. In fact, if we set R as an identity matrix (identity matrix is a rotation matrix), RS’R’W = S’W. As shown in the table, S’W is much better than random RS’R’W. It seems that **the selection of the rotation matrix R is critical to the final segmentation performance**.
> > > > 2. If we consider RS’R’ (it is a diagonal matrix in the initialization stage) as a whole, RS’R is only related to one dimension of the weight W. Thus for the middle matrix S’, **it is also channel-aligned with respect to weight W**.
> > > > 3. But if R is random initialized, **we can not guarantee that RS’R’ is a diagonal matrix when updating S’ during training** (we verify this phenomenon with the saved checkpoints when we finish the training). Note that the weight W is the one from the pre-trained backbone, which contains semantic clues or learned knowledge. The non-diagonal matrix RS’R’ may bring unexpected transformation to the pre-trained weight W, leading to poor results.
> > > >
> > > > In addition, following the reviewer's suggestion, we upload a new revision supplementary, where we add these results and analysis in above discussion to supplementary D.3 (marked in red).

---

> > > > > ### Comment · Reviewer_8FCE · 2022-08-06
> > > > > **Fascinating update**
> > > > >
> > > > > Thank you for the impressive response time and the new results, this is a fascinating and (to me) very surprising outcome! Regarding the explanations, I don’t think 2 and 3 hold – if we let R=U, then according to arguments 2 and 3 this model should still underperform, but in this case the model reduces to: RS’R’W -> US’U’W -> US’U’USV -> US’SV. This is functionally equivalent to USS’V, which we know performs quite well. Nevertheless, these new results pretty conclusively demonstrate that channel-aligned fine-tuning is not uniquely destructive; rather the singular value space is indeed uniquely non-destructive, as originally claimed. I’d highly recommend repeating this experiment (perhaps using URS’R’V for the absolutely most rigorous comparison) and adding it to the main paper, possibly as an extension to Tab.7, as it greatly strengthens the argument being advanced. Regardless, I’ve raised my rating to Accept – I still don’t know what makes the singular value space so special, but it matters less in the face of such a strong empirical argument that it truly is so.

---

> > > > > > ### Author Response · Authors · 2022-08-09
> > > > > > **Response to reviewer 8FCE**
> > > > > >
> > > > > > We agree with the reviewer that good performance should be achieved when R=U. The above analysis of 2 and 3 is for random rotation matrix, and does not include the special case of R=U. According to the above results and analysis, we conclude that the choice of R is very important.
> > > > > >
> > > > > > Following the reviewer's suggestion, we conduct two experiment for the absolutely most rigorous comparison, where the weight W becomes URSR'V$^T$ (Fine-tuning S or freeze backbone). The results below show that introducing a random rotation matrix R gives poor results. It demonstrate that the introduction of random rotation matrices (without R=U and R=I) destroys semantic clues in pre-train weights. Meanwhile, we find that fine-tuning the singular value space S brings positive effects to the model under different weights. It proves that the singular value space is indeed uniquely non-destructive.
> > > > > >
> > > > > > | Mehod  | Backbone  |Expression of weight |Fine-tune param | Fold-0  |  Fold-1 | Fold-2  | Fold-3  |  Mean |
> > > > > > | ------------ | ------------ | ------------ | ------------ | ------------ | ------------ | ------------ | ------------ | ------------ |
> > > > > > | baseline  | ResNet-50  | W  | -  | 65.60| 70.28| 64.12| 60.27| 65.07 |
> > > > > > |  baseline |ResNet-50  |USV$^T$|S|  67.42  | 71.57  | 67.99  | 61.57  | 67.14  |
> > > > > > |  baseline |ResNet-50  |URSR'V$^T$|S|  23.20  | 35.62  | 34.52  | 27.69  | 30.26  |
> > > > > > |  baseline |ResNet-50  |URSR'V$^T$|-|  23.64  | 33.19  | 33.89  | 26.51  | 29.31  |
> > > > > >
> > > > > > In addition, we add this experiment to main paper page 9  Section 4.4 & Tab.9 (marked in red).

---

### Official Review · Reviewer_xj2U · 2022-07-12

**Rating:** 6
**Confidence:** 4
**Soundness:** 3 good
**Presentation:** 3 good
**Contribution:** 3 good

**Summary:**

In order to handle the overfitting issue in few-shot segmentation, this paper proposes to fine-tune a small part of backbone parameters recognized via the singular value decomposition. Precisely, the proposed Singular Value Fine-tuning (SVF) method suggests merely tuning the decomposed singular-value diagonal matrix for each convolutional layer. The experiments on few-shot segmentation among two datasets show the positive effects of fine-tuning the backbone using the SVF approach.

**Questions:**

The major concern is the vague motivation since the reviewer cannot find any connection of the proposed Singular Value Fine-tuning is specially designed for the “few-shot” scenario. In addition, the other concern of this paper is lacking compared with previous methods [A, B] related to manipulating the weight matrices of the convolutional layers. Please see [Weaknesses] for reference.

**Limitations:**

The authors adequately addressed the limitations and potential negative societal impact of their work.

**Strengths And Weaknesses:**

[Strengths]
+ The idea of recognizing the singular-value-related backbone parameters for fine-tuning is interesting.
+ The manuscript is well organized and has several experiments.

[Weaknesses]
- The motivation is vague since the strong connection between the backbone fine-tuning and few-shot segmentation is unclear. The frozen pretrained-backbone could be treated as a feature extractor for a downstream task; hence the downstream task could focus on the mechanism design for employing the extracted features. For fine-tuning a backbone network as a goal, why not compare with the methods of meta-learning, adaptor, bias tuning, or domain adaption? Therefore, it is curious why it must fine-tune the backbone for tackling the few-shot segmentation. Specifically, the reviewer cannot find any components within the proposed Singular Value Fine-tuning specially designed for the “few-shot” scenario. As a result, it is suggested to deploy the proposed Singular Value Fine-tuning on various tasks, including classification and detection, to demonstrate the overfitting-proof advantages of the proposed SVF method.
- In line 78, the authors claimed that the work [10] is not suitable for few-shot vision tasks, yet there are no related experiments showing this issue.
- Since the proposed SVF is used to decompose the basic convolutional layers within a backbone, it may result in additional computational time while carrying out SVF. Therefore, it is better to include the experiments in discussing the required extra training time.
- The reference is not sufficient. For example, two related methods shown as follows should be compared.
[A] Kui Jia, Dacheng Tao, Shenghua Gao, Xiangmin Xu: Improving Training of Deep Neural Networks via Singular Value Bounding. CVPR 2017: 3994-4002.
[B] Hanie Sedghi, Vineet Gupta, Philip M. Long: The Singular Values of Convolutional Layers. ICLR 2019.

---

> ### Author Response · Authors · 2022-08-02
> **Response to reviewer xj2U (1/2)**
>
> Thanks a lot for your time and feedback. We given the responses to all raised concerns below.
>
> ___
>
> **Q1: The strong connection between the backbone fine-tuning and few-shot segmentation is unclear. It is curious Why it must fine-tune the backbone for tackling the few-shot segmentation.**
>
> **A1:** There may be some misunderstandings, and we provide further explanations about our SVF in this response. We did not say it *"must"* fine-tune the backbone for few-shot segmentation. Instead, we agree with the claim that freezing the backbone in few-shot segmentation is a good way to achieve promising segmentation results. In this paper, we revisit the above conventional paradigm and provide **an alternative way** -- fine-tuning a small part of parameters in the backbone. And the experimental results show that the alternative regime can achieve better results on various few-shot segmentation methods over the conventional paradigm. Thus, **the connection** between the backbone fine-tuning and few-shot segmentation lies in: **fine-tuning part of parameters in the backbone can serve as an alternative way to the freezing backbone paradigm in few-shot segmentation and can give non-trivial improvements over various few-shot segmentation methods.** Our method brings new thoughts to few-shot segmentation. It suggests that not just the mechanism design in fusing different extracted features or generating prototypes is essential, but **the quality of the extracted features from the backbone also matters** to the final segmentation results.
> ___
>
> **Q2: For fine-tuning a backbone network as a goal, why not compare with the methods of meta-learning, adaptor, bias tuning, or domain adaptation?**
>
> **A2:** Thanks for your constructive suggestions of comparing our SVF with Adapter and Bias Tuning. For quick check, we conduct experiments on Pascal-5$^i$ with the 1-shot setting. The details for adapter and bias tuning are given below:
>
> - Adapter: Adapter is proposed in transformer-based models. When applying it into CNN-based backbone (ResNet), we make simple adjustments. We follow [C] to build the adapter structures and add them after the stages in the ResNet.
> - Bias Tuning: In the ResNet backbone, the convolution layers do not contain bias term. The bias terms that can be used for tuning is the ones in BN layers. We fine-tune the bias terms in all BN layers in this method.
>
> The experimental results are given in the table below. It shows that **SVF outperform Adapter and Bias Tuning by large margins**. Moreover, we find that the introduction of Adapter will directly lead to over-fitting, while Bias Tuning reduces performance of the baseline model.
>
> | Method  | fine-tune method  | Fold-0  | Fold-1  | Fold-2  | Fold-3  | Mean  |
> | ------------ | ------------ | ------------ | ------------ | ------------ | ------------ | ------------ |
> | baseline  |  Freeze Backbone | 65.60  | 70.28  | 64.12  | 60.27  | 65.07  |
> | baseline  |  SVF | 67.42  | 71.57  | 67.99  | 61.57  | **67.14**  |
> | baseline  |  Adapter | 18.41  | 20.21  | 26.62  | 17.62  | 20.71  |
> |  baseline |  Bias-Tuning |  61.62 |  70.10 | 64.80  | 55.19  | 62.93  |
>
> For meta-learning and domain adaptation, we would like to make some clarifications.
>
> - In the few-shot segmentation, meta-learning is applied in the segmentation head to learn the knowledge in support images but not in the backbone, posing challenges in directly comparing SVF with meta-learning methods.
> - In addition, domain adaptation is another research direction whose setting differs from the setting in few-shot segmentation. It would be much appreciated if the reviewer could give more details on conducting fair comparisons between our SVF and domain adaptation methods.

---

> > ### Author Response · Authors · 2022-08-02
> > **Response to reviewer xj2U (2/2)**
> >
> > **Q3: The reviewer cannot find any components within the proposed SVF specially designed for the "few-shot" scenario. As a result, it is suggested to deploy the proposed SVF on various tasks, including classification and detection, to demonstrate the overfitting-proof advantages of the proposed SVF method.**
> >
> > **A3:** We understand the reviewer's concern and would like to provide more explanation about our logic:
> >
> > - First of all, **this paper focuses on the few-shot segmentation task**.
> > - Then, we notice that current few-shot segmentation methods follow a paradigm of freezing backbone. By revisiting this paradigm, we find this convention exists may due to the fact that fine-tuning backbone results in overfitting [D,E].
> > - To solve this problem, we provide SVF as a solution, which only fine-tunes part of parameters in the backbone and gives better results.
> >
> > Thus, **SVF is proposed to solve the existing problem in few-shot segmentation**.
> >
> > Moreover, we would like to thank the reviewer for recognizing our SVF as a general method that can be applied to few-shot classification and few-shot object detection. As the settings and baseline methods may change in the above two tasks, applying SVF to these two tasks need more specific designs according to task settings. We leave them for future work.
> >
> > ___
> >
> > **Q4: In line 78, the authors claimed that the work [10] is not suitable for few-shot vision tasks, yet there are no related experiments showing this issue.**
> >
> > **A4:** Thank you for pointing this out. We notice that the claim in Line-78 is not appropriate. We revise it here: The above methods are proposed in a transformer-based model, but modern few-shot segmentation models use CNN-based backbones. Applying prompt-based methods to various few-shot segmentation methods may need further adjustments. We have added this new sentence to the revised version of our paper.
> >
> > ___
> >
> > **Q5: It is better to include the experiments in discussing the required extra training time.**
> >
> > **A5:** We follow the reviewer's advice and measure the training time of models on Pascal-5$^i$ with the 1-shot setting. Compared with the baseline model (freeze backbone), SVF increases the training time from 2 hours to 5.5 hours on Fold-0. Given the setting of the few-shot scenario, there are only limited samples, enabling fast training for models. It is acceptable even if the training time increases. Moreover, SVF is only applied in model training and does not affect model inference (in inference, we combine the U, S, and V back to the weight of convolution layers, which is the same as the original model).
> >
> > ___
> >
> > **Q6: The reference is not sufficient.**
> >
> > **A6:** Thanks for pointing it out. We will include all those works in our final version. The detailed discussions are as follows:
> >
> > - Both [A] and [B] constrain the distribution of the singular values $s$ where [A] forces the singular value around 1 and [B] clamps the large singular values into a constant, hence serving as a regularization term. We did not pose an extra constraint on $s$, instead, encouraged the fully trainable singular values.
> > - As illustrated in [A]'s Figure 1, the singular values of well-trained weights are widely spread around [0,2]. The strong regularization proposed in [A,B] should damage the performance of pre-trained networks. Therefore, they turn to training from scratch, which is infeasible in the circumstance of few-shot segmentation. Our method coupled with pre-trained parameters can further exploit the capacity of the backbone, leading to superior results.
> >
> > ___
> >
> > [A] Kui Jia, Dacheng Tao, Shenghua Gao, Xiangmin Xu: Improving Training of Deep Neural Networks via Singular Value Bounding. CVPR 2017: 3994-4002.
> >
> > [B] Hanie Sedghi, Vineet Gupta, Philip M. Long: The Singular Values of Convolutional Layers. ICLR 2019.
> >
> > [C] Houlsby, Neil, et al. Parameter-efficient transfer learning for NLP. ICML. PMLR, 2019.
> >
> > [D] Dong, Nanqing, Eric P. Xing. Few-shot semantic segmentation with prototype learning. BMVC. Vol. 3. No. 4. 2018.
> >
> > [E] Min, Juhong and Kang, Dahyun, Cho, Minsu: Hypercorrelation squeeze for few-shot segmentation. ICCV 2021: 6941--6952

---

> > > ### Comment · Reviewer_xj2U · 2022-08-08
> > > **Thanks for the authors' response**
> > >
> > > Thanks for the authors' response. Since the authors addressed my initial concerns about the backbone fine-tuning, more comparisons, extra training time, and more references, I would like to raise my final rating.

---

> ### Author Response · Authors · 2022-08-06
> **Follow-up questions:**
>
>
> We appreciate your valuable comments. We were wondering if our responses have addressed your concerns. Please let us know if you have additional questions. Thank you!

---

### Meta-Review · Area_Chair_Fhbf · 2022-08-26

**Recommendation:** Accept
**Confidence:** Certain

**Metareview:**

This paper presents a solution to overcome the overfitting problem in few-shot segmentation. Specifically, the proposed method decomposes the backbone parameters into three matrices via singular value decomposition (SVD) and fine-tunes only the singular values, while leaving the others frozen. This allows the model to adjust the feature representation in a new class while maintaining the semantic cues in the pre-trained backbone. All reviewers admit that this paper is well written, and the proposed method is applicable and novel. Furthermore, the authors provide great additional experiments and answers to the reviewers’ concerns. These made all reviewers positive for this paper. The AC agreed with the reviewers that the proposed method would make waves in the few-shot learning paradigm where the parameters of the pre-train model should be frozen. The AC recommends including the results described in the rebuttal for the final camera-ready version.


**Award:**

No

---

### Decision · Program_Chairs · 2022-09-14

Accept